# Morphogenesis of starfish polymersomes

Jiawei Sun[1], Sandra Kleuskens[1], Jiabin Luan [1], Danni Wang [1], Shaohua Zhang [1], Wei Li [1], Gizem Uysal [1] & Daniela A. Wilson [1] ✉

The enhanced membrane stability and chemical versatility of polymeric vesicles have made them promising tools in micro/nanoreactors, drug delivery, cell mimicking, etc. However, shape control over polymersomes remains a challenge and has restricted their full potential. Here we show that local curvature formation on the polymeric membrane can be controlled by applying poly(*N*-isopropylacrylamide) as a responsive hydrophobic unit, while adding salt ions to modulate the properties of poly(*N*-isopropylacrylamide) and its interaction with the polymeric membrane. Polymersomes with multiple arms are fabricated, and the number of arms could be tuned by salt concentration. Furthermore, the salt ions are shown to have a thermodynamic effect on the insertion of poly(*N*-isopropylacrylamide) into the polymeric membrane. This controlled shape transformation can provide evidence for studying the role of salt ions in curvature formation on polymeric membranes and biomembranes. Moreover, potential stimuli-responsive non-spherical polymersomes can be good candidates for various applications, especially in nanomedicine.

As one of the nature inspired membrane-based structures, polymersomes have gained much attention in recent years, along with new advances in controlled polymer synthesis[1]. Their enhanced membrane stiffness and versatility in terms of flexibility and chemical design of the membrane and its properties have made them promising tools in the fields of micro/nanoreactors, drug delivery, cell mimicking, etc[2–4]. However, unlike the inorganic nanoparticle synthesis, where non-spherical nanoparticles can be routinely produced[5], the scant shapes obtained from polymersomes restricted their full potential. In nanomedicine, the shape of nanocarriers is considered a new parameter in governing their performance in vivo, as crucial as polymer property, particle size, and surface chemistry[6,7]. Moreover, a successful cell mimicking for elucidating underlying mechanisms of the formation of non-spherical biomembrane shapes also requires more control over shapes of polymersomes[8,9]. Shape control methods promise to generate polymersomes with different morphologies and perform shape transformations on the bilayer membranes which are not yet available nevertheless expected from scientists in different fields[10,11].

To address this issue, several strategies have been applied over the past decades. By applying osmotic pressure to polymersome membrane through dialysis[12], salt addition[13,14], and poly(ethylene glycol) PEG addition[15], researchers were able to form stomatocytes (bowl-like polymersomes), tubes and stomatocyte-in-stomatocyte, etc.

Another strategy is to introduce liquid-crystalline (LC) side chains into block copolymers and induce shape transformation based on the confinement of a two-dimensional LC lattice that is assembled within the polymeric shell. Based on this strategy, Li et al.[16,17] and Thordarson et al.[18,19] were able to make ellipsoidal polymersomes, faceted polymersomes, and tetrahedral polymersomes. In comparison, liposomes have been observed to form much more different morphologies from not only the response to environmental changes like osmotic pressure and temperature[20,21], but also the protein interactions inspired by biomembranes[22–26]. Membrane proteins can induce shape transformation in cells when interacting with biomembrane, in order to mimic this shape transformation, a strategy has been developed recently in our group, in which poly(*N*-isopropylacrylamide) (PNIPAm) was used as alternatives for proteins with a hydrophobic unit to insert into polymersome membrane[27]. The study proved that membrane insertion is a successful polymeric-based shape transformation strategy. Several polymersome-based non-axisymmetric shapes were fabricated, such as polymersomes with tentacles, boomerang-like, rackets-like shapes etc. PNIPAm formed particles through interchain aggregation during the process, similar to the aggregation of proteins, the self-assembly of surfactants above critical micelle concentration, or any other amphiphilic substances. This aggregation could lead to different membrane interactions and result in different curvatures[25,28]. In

[1]Institute for Molecules and Materials, Radboud University, Heyendaalseweg 135, 6525 AJ Nijmegen, the Netherlands. ✉e-mail: d.wilson@science.ru.nl

nature, ions as endogenous small molecules play specific roles in various biological functionalities. For example, ion interaction was found to influence the stability of proteins, which can further affect their physiological signalling function[29]. To study the controlling of membrane proteins and membrane transformation using salt ions can be meaningful for a better understanding of the natural membrane. Interestingly, ions can also change the properties of polymers like PNIPAm. The lower critical solution temperature (LCST) of PNIPAm can be decreased when interacting with the ions added in the solution[30–33]. Moreover, ions can also provide extra osmotic pressure against the polymeric membrane, resulting in a minor volume reduction that leads to a different shape transformation[34,35].

In this study, we designed a method based on the membrane insertion strategy to further explore the morphogenesis of polymeric vesicles. In the meantime, the importance of salt ions in the shape transformation of polymeric particles is also clarified. PNIPAm was used as a hydrophobic motif in polymeric membrane insertion, and salt ions were added into the system to tune the property of PNIPAm and its insertion behaviour. This strategy changed polymersomes to multi-armed starfish with the number of arms depending on the salt concentration. The mechanism behind the multi-arm formation is studied, the insertion of PNIPAm was visualized using the negative staining method and TEM-energy dispersive X-ray mapping analysis. Moreover, octopus-like shapes were also observed while increasing the membrane flexibility. These results could provide evidence to support the role of salt ions in the formation of curvature in polymeric membranes and understand natural processes where salt ions and membrane proteins play crucial roles. Meanwhile, the shapes observed could be a good candidate for designing stimuli-responsive nanomotors.

## Results and discussion
### Curvature formation of polymersomes via PNIPAm insertion assisted by salt ions

A previous study conducted by our group demonstrated that external hydrophobic polymers inserted in the membrane can induce local membrane curvature[20,27], and volume reduction is also an important factor in shape transformation[35]. The addition of salt ions during curvature formation might result in shapes with small reduction volume due to the extra osmotic pressure experienced by polymeric membrane[35]. In nature, these salt ions play essential roles in manipulating biomembrane proteins. Whether this ion interaction would occur on the membrane protein mimetic polymer PNIPAm raised our interests. To explore the potential impact on shape transformation, we first studied salt-ion-induced volume reduction by introducing different amounts of $NaNO_3$, ranging from 1 to 60 mM, into the polymersome solutions. To make the membrane flexible for further shape changing, organic solvent tetrahydrofuran (THF) and dioxane (4:1) were added to the solution through slow addition, as they have a plasticizing effect on polystyrene. After adding 23% organic solvent (THF and dioxane), the samples were quenched and analysed using an electron microscope (EM) as depicted in Fig. 1a1–e1. In the absence of salt ions, the polymersomes can be transformed into flat disks (Fig. 1a1, a2, g). The addition of salt ions further drove the shape transformation towards half-open stomatocytes-like polymersomes (Fig. 1h) due to the additional reduction in inner volume. This reduction is evident from the decreased distance between two bilayers in the EM images. In addition, no other non-axisymmetric shapes were observed, indicating that salt addition alone may not play a role in the membrane composition change[27,34]. To quantitatively describe the volume reduction, all half-open stomatocytes obtained by cryo-TEM were fitted using a parameterization based on a Fourier series[36] (Fig. 1a2–e2). The reduced volume ($\Delta v$) and reduced area difference ($\Delta a$) between the outer and inner layers of the membrane were calculated to construct a phase diagram (Fig. 1f, Supplementary Fig. 1 and Supplementary Table 1).

When polymersomes were transformed to flat disks, $\Delta v$ was reduced by 40%[36]. When different concentrations of $NaNO_3$ (1 mM, 3 mM, 15 mM and 60 mM) were added to the system, the $\Delta v$ decreased by 49%, 62%, 64% and 66%, respectively, compared to 0 mM salt (Fig. 1f). These findings suggest that the addition of salt ions can decrease the inner volume of polymersomes, as the polymersome membrane is subjected to additional osmotic pressure[37]. Besides, this osmotic pressure adds energy to the system which can stored on the polymersome membrane in the form of bending energy, results into deformation of the polymersome. For small inner volume polymersome, the high bending energy inhibits further volume reduction as more osmotic pressure is needed. (Supplementary Fig 1). The reduction in inner volume had almost no effect on shape transformation, indicating that tuning only the volume reduction is insufficient for further shape transformation. Although volume reduction has been considered the driving force in the formation of structures like multi-armed starfish etc in liposome systems[34,35,38], these results demonstrate that it alone does not lead to additional shape transformation. As studied in our previous work[27], membrane composition change was considered the critical condition for shape changing. We, therefore, assume that membrane composition change could be the main factor in the formation of arms. To prove this hypothesis, 250 µg PNIPAm was added to the solution together with different amounts of $NaNO_3$. After 23% organic solvent (THF and dioxane) was added to the solutions, the samples were quenched and examined with TEM (Fig. 1i1–n1) and Cryo-SEM (Fig. 1i2–n2). Since PNIPAm can exhibit a change in property from hydrophilic to hydrophobic when a certain amount of organic solvent (THF, Methanol, etc.) is added to its solution[39], we should expect PNIPAm to insert into polymeric membranes as a hydrophobic motif and change the membrane composition based on hydrophobic effect[27]. With the increase of salt concentration, polymersomes changed from boomerang-like shapes, tubes, and three armed shapes to shapes with multiple arms. The higher the salt concentration, the more arms formed on polymersomes. When salt concentration reached 15 mM, 6–7 long arms were observed (Fig. 1i1, i2). When salt concentration reached 30 mM, 7–8 short arms were observed (Fig. 1m1, m2), and the number of arms could be above 10 when 60 mM salt was added (Fig. 1n1, n2). The most striking feature of these starfish shapes is their construction. As shown in the EM images, the centre consists of a flat, nearly axisymmetric core, and cylindrical arms end with spherical caps were growing out from the core[35] As the number of arms increased, the cylindrical part of the arms became shorter, likely due to a decrease in the inner volume as the cylindrical part provides more volume than the flat core. Interestingly, when 250, 500, 1000 µg of PNIPAm were added to the solution (Supplementary Fig. 2), polymersomes from all samples changed to starfish-like shapes with mostly 5 arms. This indicates that the growth of arms was not controlled by the PNIPAm concentration when it reached saturation (above 250 µg), similar to the shape changing without salt[27]. Conversely, decreasing the amount of PNIPAm resulted in a decline in the number of arms, and the increase of salt ions led to smaller arms (Supplementary Fig. 3), consistent with the observations above. These results indicate that multi-armed starfish tend to form in larger quantities when a sufficient amount of salt ions are added to the polymer membrane along with PNIPAm. Therefore, the growth of the arms may be due to the insertion of PNIPAm and the additional internal volume reduction caused by the salt ions. A control experiment was conducted to confirm whether volume reduction and PNIPAm insertion are the primary reasons for the observed shape changing. In this experiment, 120 mM of sucrose was added to apply the same osmotic pressure on the membrane, so that the same extra volume reduction can be created during solvent addition. Supplementary Fig. 4a shows that sucrose can provide a similar influence as the salt ions in reducing the inner volume. However, their effects on PNIPAm-induced protrusion formation were not the same (Supplementary Fig. 4d). As a result, sucrose could not assist

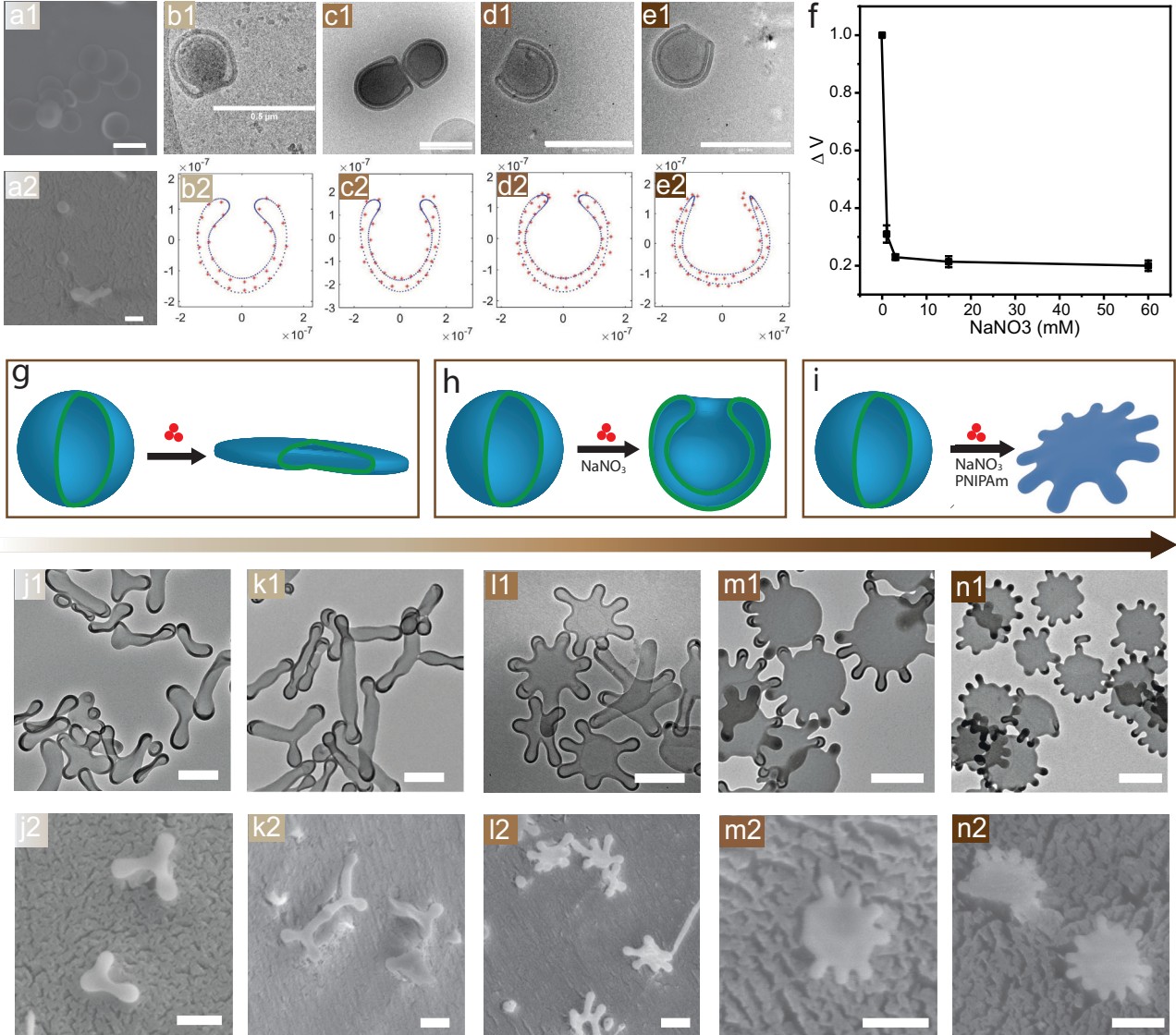

**Fig. 1 | Salt ion induced shape transformation.** SEM (**a1**) and Cryo-SEM (**a2**) of polymersomes after 23% of organic solvent (THF and dioxane) added. Cryo-TEM (**b1**–**e1**) images depicting the shape transformation of polymersomes with different amount of $NaNO_3$ (1 mM, 3 mM, 15 mM and 60 mM, respectively) added in the solutions, samples were quenched after 23% of organic solvent (THF and dioxane) were added dropwise. **b2**–**e2** Parameterization of different shapes from cryo-TEM images (unit = metre). For every fitted shape, the reduced volume (Δv) is calculated.

For each shape, 5 samples are used for calculation, as shown in Supplementary Fig. 1. **f** The correlation between reduced volume and salt concentration, error bars represent the standard deviation of Δv (*n* = 5). Illustration of polymersomes shape transformation after adding solvent (**g**), together with salt (**h**) and both salt and PNIPAm (**i**). **j1**–**n1** TEM and (**j2**–**n2**) TEM images depicting the shape transformation of polymersomes with the addition of PNIPAm (250 μg) and $NaNO_3$ (0 mM, 3 mM, 15 mM, 30 mM and 60 mM, respectively). Scale bar = 500 nm.

the membrane in forming more local curvatures. Furthermore, we investigated the mechanism behind the formation of arms by looking at the effects of salt ions on the property of PNIPAm and its insertion into the polymeric membrane.

The influence of salt ions on the properties of PNIPAm solution was observed, resulting in a decrease or increase in critical solution temperature (LCST) based on their interactions with the added ions[30–33]. This phenomenon is referred to as the Hofmeister effect, which has been widely studied in proteins. The specific interactions between salt ions and water are believed to be crucial for protein stability, denaturation and enzymatic activities[30]. Ions such as fluorides or sulfates, which are strongly hydrated, are called kosmotropes. They can steal water from the solute and gather several layers of water molecules around themselves, leading to a salting-out effect. In contrast, chaotropes ions such as perchlorates or thiocyanates, are weakly hydrated and cannot retain water around them, resulting in a salting-in

effect[40,41]. Further research also provides evidence that the interaction between ions and water should not be the only reason for the salting-out/in behaviour. Their interactions with protein surfaces have been carried out to better explain the Hofmeister effect; both the backbone and side-chain of protein have been found to be involved[42–44]. In our study, as shown in Supplementary Fig. 5, we conducted control experiments to further explore ion influence on membrane curvature formation with both the strong Hofmeister kosmotropes and chaotropes. Interestingly, strong chaotropes ions such as NaSCN showed a positive influence in the construction of membrane curvature, with an average of 11 arms formed, even slightly higher than $NaNO_3$. In addition, NaCl, which has a similar Hofmeister effect as $NaNO_3$, showed a similar impact on membrane curvature formation. On the other hand, when kosmotropes ions are added in the solution, the formation of membrane curvature is not as much as when $NaNO_3$ was added. Two kosmotropes ions $Na_2HPO_4$ and $Na_2SO_4$, were added separately in

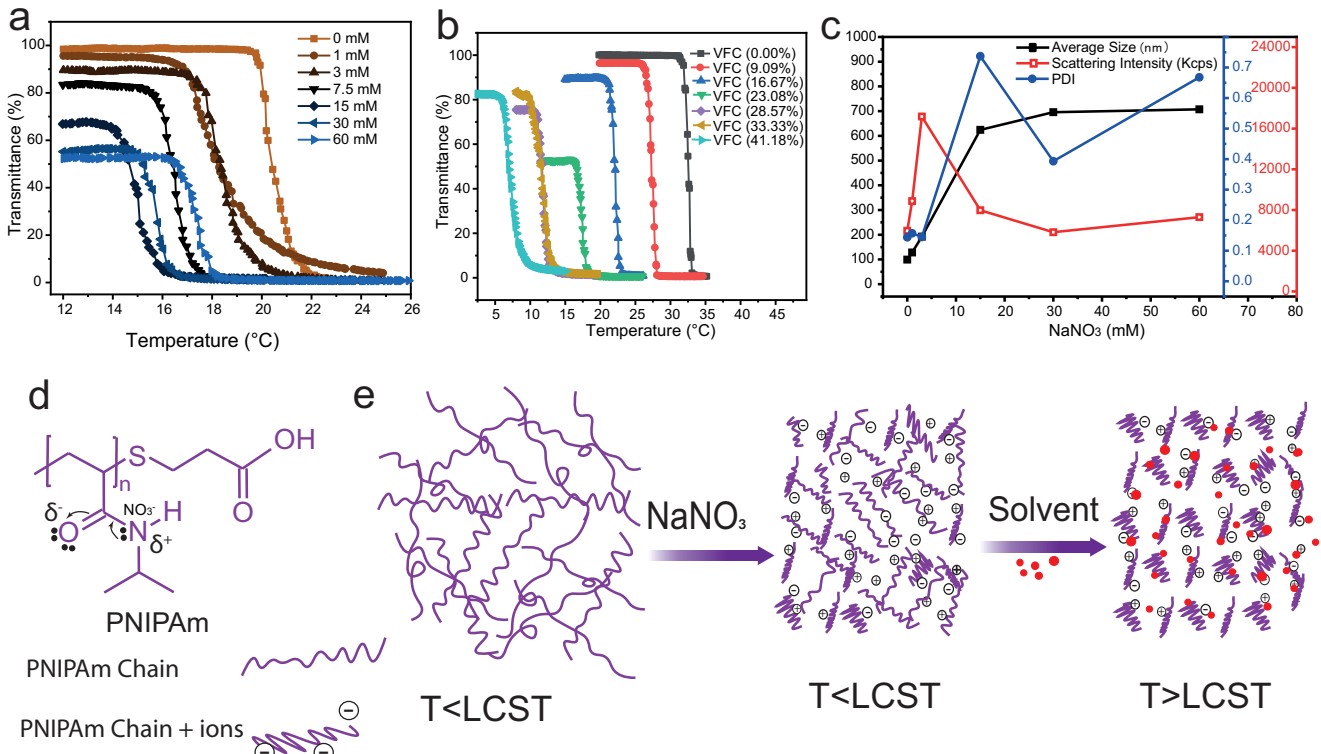

**Fig. 2 | The co-nonsolvency phenomenon of PNIPAm at different salt concentrations with VFC variate from 0 to 41%. a** PNIPAm transmittance measurement by UV-vis spectroscopy to detect the shift of LCST (lower critical solution temperature) with different salt concentrations (0–60 mM) at 23% of VFC (volume fraction of co-nonsolvent). **b** The shift of LCST with VFC varies from 0 to 41%. **c** Dynamic light scattering measurements to observe the hydrophobic behaviour of PNIPAm at the VFC of 23% with the addition of 0, 1, 3, 15, 30, 60 mM NaNO₃ in PNIPAm water solution. **d** Chemical structure of PNIPAm and possible salt interaction position between PNIPAm and salt ions. When Salt is added to PNIPAm solution, anion (nitrate ions) can directly bind to the amide group of PNIPAm, causing the destabilization through polarization. **e** Schematic representation of hydrophobic behaviour of PNIPAm chains at the addition of salt ions and solvent, salt-only cannot turn PNIPAm to hydrophobic, but after addition of organic solvent (THF and dioxane), PNIPAm became hydrophobic, however, no interchain aggregation was observed. PDI indicates the polydispersity index of the samples.

these experiments, and both of them were found less effective than the chaotropes. Na₂SO₄, in particular, showed a significantly weaker influence in the formation of curvatures, with only 3−5 arms were formed on average. This could be due to the fact that chaotropes are capable of penetrating the hydrophilic region of a hydrocarbon-packed monolayer and even lipid membranes, as the presence of chaotropic salts in the subphase increased the surface pressure[45,46]. We believe the chaotropic salts in our experiment had a similar effect on the polymeric membrane, the presence of chaotropic salt ions has facilitated the PNIPAm insertion comparing to the presence of kosmotropic ions.

## Salt ions effect on PNIPAm property

In order to investigate this possible interaction between salt and PNIPAm, we first evaluated the conformational change of PNIPAm by measuring solution transmittance using UV-vis spectroscopy at different NaNO₃ concentrations (0–60 mM) when 23% of volume fraction of co-nonsolvent (VFC) was added into the PNIPAm water solutions (Fig. 2a, b). The LCST of PNIPAm decreased when salt concentration increased from 0 mM to 15 mM. However, when more NaNO₃ was added into the system, the LCST slightly rose back. The rise will be explained later. Another interesting observation was that the specific ion interaction with PNIPAm at 23% of VFC showcases a hydrophobic change beyond LCST. This means that even at low temperature, PNIPAm was slightly hydrophobic as seen from the decrease of transmittance in Fig. 2a. The percentage of transmittance at a lower temperature (T < LCST) was dependent on salt concentration, the higher

the salt concentration, the lower the transmittance is. This indicates that the salt-induced conformational change in PNIPAm is different from that caused by an organic solvent. This has been studied in Previous reseach[27,47,48], the LCST of PNIPAm can be decreased by co-nonsolvency phenomenon, therefore with the change of solvent ratio, LCST changes. Simply put, the phase behaviour of PNIPAm in the water-rich and the alcohol-rich regime is different. Adding solvent to water decreases the enthalpy of the bulk water due to the kosmotropic effect. In this experiment, the effect suppresses the hydrophobic hydration between water and hydrophobic units of the PNIPAm while adding water to alcohol decreases the solubility of PNIPAm in the solvent mixture[49]. On the other hand, as a weak chaotrope, NaNO₃ can have a salt-in effect on PNIPAm as studied previously[50], meaning that they could stabilize PNIPAm by binding directly to the polyamide in a salt concentration-dependent manner. Since the organic solvent is also involved in our study, PNIPAm is dispersed in the quaternary PNIPAM/water/solvent/salt system. In this system, both solvents and salt are agents that effectively affect the PNIPAm, assist each other to increase the surface tension of the cavity surrounding the backbone and the isopropyl side chains[50−52], so that the LCST slightly decreased as showing in Fig. 2a. When the salt concentration is higher than 30 mM, the ions that break the hydrophobic hydration (water structure around hydrophobic molecules) become saturated, and extra ions may compete to bind on the side-chain amide moieties, leading to a salt-in effect[33,50,53], which increases LCST (Fig. 2a). On the other hand, the cancellation of LCST may be due to the

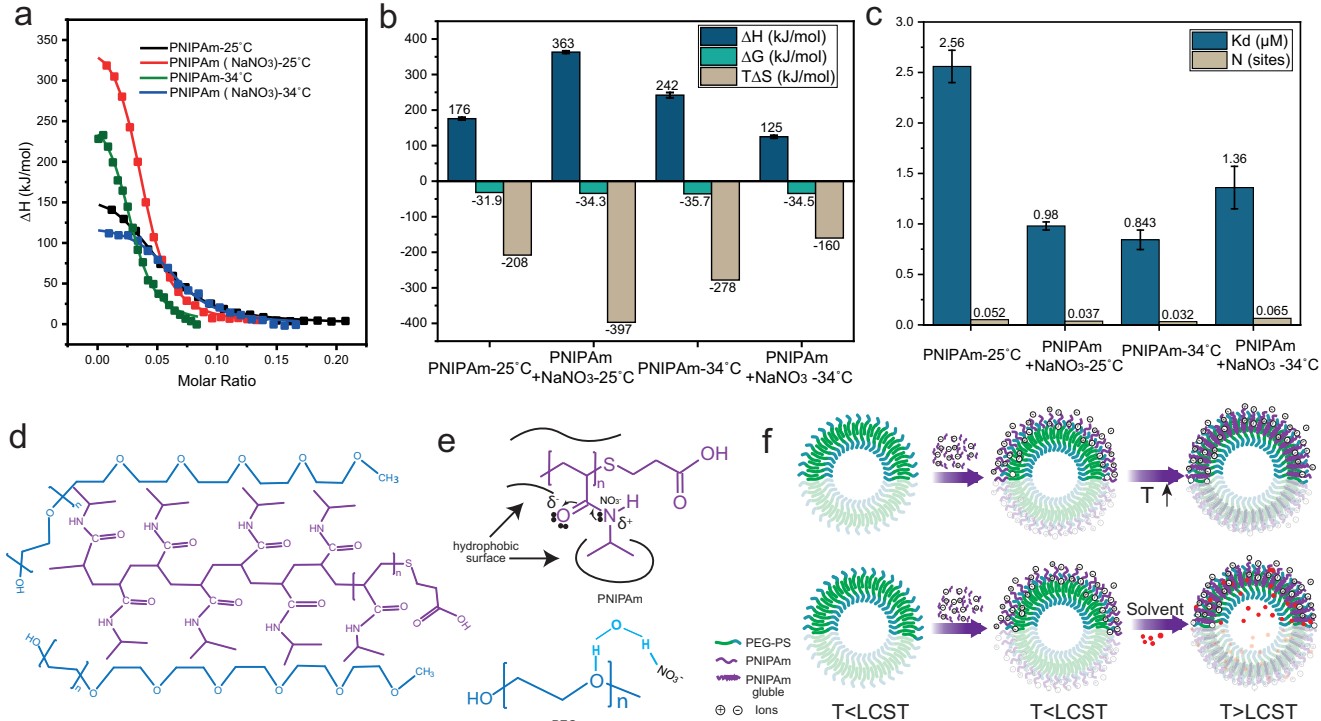

**Fig. 3 | Isothermal titration calorimetry (ITC) measurement for the association between PNIPAm and polymeric membrane. a** Endothermic titration thermograms integrated from titrating PNIAPm into 2.5 mM polymeric vesicle solutions at 25 ˚C, 34 ˚C, and when 10 mM NaNO$_3$ was added into the two groups. **b** ΔG, ΔH and −TΔS from ITC measurements of the 4 groups. **c** Binding affinity ($K_d$) and binding sites (N) from ITC measurements, error bars indicate the errors of integrated heats and estimated $K_d$. **d** Illustration for the possible interaction between PEG corona (blue) and PNIPAm (purple), PNIPAm is inserted between PEG chains by hydrophobic effect. **e** The hydrophobic hydration of the PNIPAm is associated with surface tension and can be modulated by anions; Anions can bind to the amide group of PNIPAm directly; NO$_3^-$ can destabilize hydrogen bonding between the ester oxygen and water molecules through polarization. **f** Schematic representation of PNIPAm polymeric membrane interactions with the presence of salt, salt slightly salted out PNIPAm, assisted the hydrophobic association between PNIPAm and polymersomes while increasing temperature (top), addition of organic solvent would also turn PNIPAm to hydrophobic and result in the same hydrophobic association as temperature increasing.

formation of a poor solvent system, where hydrophobic hydration is destroyed, resulting in partially hydrophobic PNIPAm.Amide dehydration could be uncoupled from the rest of the phase transition, resulting in the formation of partially collapsed structures that scatter light even at very low temperature[50]. In addition, when different amounts of solvent were added to the PNIPAm with 60 mM of NaNO$_3$, the partially hydrophobic phenomenon changed accordingly. The lowest transmittance appeared when 23% of solvents were added to the system, and more solvent did not result in lower transmittance (Fig. 2b), indicating that the hydrophobic hydration in this system is solvent ratio dependent.

Overall, the co-nonsolvency phenomenon is the primary driving force behind the LCST change of PNIPAm, with salt ions affecting the LCST based on the co-nonsolvency phenomenon. This salt effect on LCST at the microscale has also been investigated using dynamic light scattering. As revealed previously, PNIPAm aggregated to nano-sized particles when 23% of organic solvent (THF and dioxane) was added to its water solution[27]. However, the particles changes with different amounts of salt ions added in the solution (Fig. 2c). For instance, when 15 mM of NaNO$_3$ is added, the size of PNIPAm particles swells from 100 to 600 nm, and the PDI increases from 0.1 to 0.7. In contrast, the scattering intensity increases when 3 mM salt is added and then decreases further when more salt is added, indicating that particles swell and disassemble through salt addition. Furthermore, when the salt concentration increases to 30 and 60 mM, the size and light scattering intensity of these samples examined with DLS

did not significantly change, indicating that no particles are formed. Although the UV-Vis experiments have confirmed the hydrophobicity of PNIPAm, DLS results reveal that the PNIPAm chains remain separated due to salt ions binding, while also becoming hydrophobic (Fig. 2e). This facilitates the insertion of PNIPAm into the polymeric membrane.

## Isothermal titration calorimetry (ITC) measurements

The observed salt influence on PNIPAm conformation does not directly prove its effects on PNIPAm and membrane interaction. To further investigate this insertion difference before and after the addition of NaNO$_3$, isothermal titration calorimetry (ITC) was used to determine the interaction change by measuring the heat absorbed or released during binding. The ITC has restrictions on the use of organic solvents. Therefore, temperature increase is used as an alternative strategy in our experiment. Its theoretical basis is that lower critical solution temperature (LCST) of PNIPAm in water is strongly related to the destabilization of hydrogen bonds between water molecules and amide groups with increasing temperature, induced by the presence of the hydrophobic isopropyl group and backbone[54,55]. The solvent mixing process at low temperatures is favoured by the formation of hydrogen bonds thermodynamically, which leads to a large negative enthalpy of mixing[56]. When a mixed solvent is added to the solution, PNIPAm solubility is reduced within a range of intermediate solvent concentrations in binary aqueous solutions. This phenomenon suggested that 'water−solvent complexes' are preferred to PNIPA−water hydrogen bonds[57]. This indicates that both LCST changes share similar chemistry. Moreover, systematic research has been done to compare

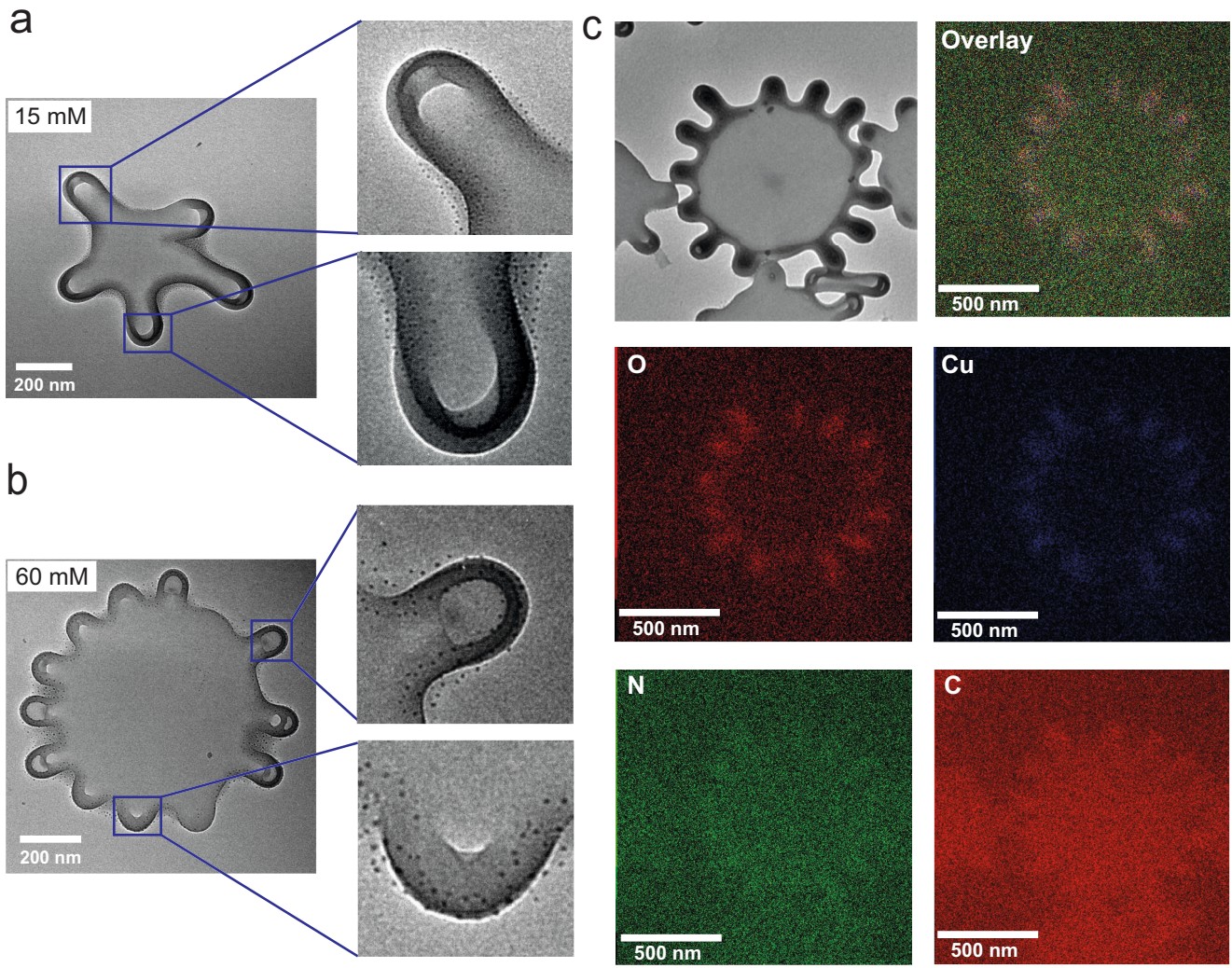

**Fig. 4 | Negative staining of polymersomes with protrusions formed among the membrane with copper sulfate (CuSO₄).** TEM image of polymersomes with 15 mM (**a**) and 60 mM (**b**) added during shape changing and stained with CuSO₄, coordination of $Cu^{2+}$ with amides from PNIPAm chains make it visible (black dots). **c** EDX element maps of polymersomes with multiple protrusions, O, Cu, N, C were detected.

these changes[58]. Using IR and micro-Raman spectroscopy, the authors showed that gradual redshifts of the C−H stretching and the amide II bands happen with increasing temperature or DMSO concentration. We believe ITC can be a good approach to assist the further exploration. The measurement principle of ITC is based on both titration and power compensation techniques. Titration calorimetry measures the enthalpy change of a chemically reacting system and a noncovalent binding event. It is widely used to elucidate the mechanisms of molecular interactions. In this experiment, the MicroCal PEAQ-ITC (automated version, Malvern analytical) was utilized to accurately determine the thermodynamic parameters, including the binding constants ($K_B$), reaction stoichiometry (n), enthalpy ($\Delta H$), and entropy ($\Delta S$). These parameters provide a complete thermodynamic profile of the different interactions ranging from exothermic to entropy-driven[59]. The experiments were conducted by titrating PNIPAm to a polymeric vesicle solution assembled from 2.5 mM PEG-b-PS, as shown in Fig. 3a, b. Four groups of heat exchange were measured, including the interaction between PNIPAm and polymeric vesicles when PNIPAm was hydrophilic (25 °C) or hydrophobic (34 °C) and when salt ions were added into the above two groups. The measurement also included a control group to observe the dilution of PNIPAm. As with any spontaneous binding process, the interaction between PNIPAm and polymeric membrane PEG corona can occur only when associated with

a negative binding free energy ($\Delta G$), which is the sum of $\Delta H$ and $-T\Delta S$. The ITC results from all four groups demonstrated that $\Delta G$ was smaller than −30 KJ/mol, indicating association in all groups. Even when measured at 25 °C, typical endothermic titration thermograms were integrated from the titration from these four experimental groups (Fig. 3a), confirming that, as expected, the hydrophobic side chains from PNIPAm can interact with the PEG corona, despite being hydrophilic (Fig. 3b). The endothermic interaction observed is attributed to the hydrophobic interaction between the PEG corona and PNIPAm (Fig. 3d). This is because dehydration of PEG from the water phase to interact with PNIPAm hydrophobic groups is an endothermic process and the process of breaking the hydrophobic hydration on PNIPAm hydrophobic residues[59–61]. During the transition from hydrophilic into hydrophobic, the interaction can facilitate further insertion. Interestingly, this association between PNIPAm and the PEG corona is much more complex than its interaction with free PEG polymers. When PNIPAm was titrated in PEG solution, an endothermal interaction was observed (Supplementary Fig. 6). The $T\Delta S$ required to break the water layers for this hydrophobic interaction is much smaller (25.4 KJ/mol) than the entropy needed to interact with the polymersome PEG corona (208 KJ/mol). This indicates that when PNIPAm is interacting with PEG corona, the interchain hydrogen bonding between PEG chains is destroyed, which requires a high energy[62]. In addition, the PEG chains

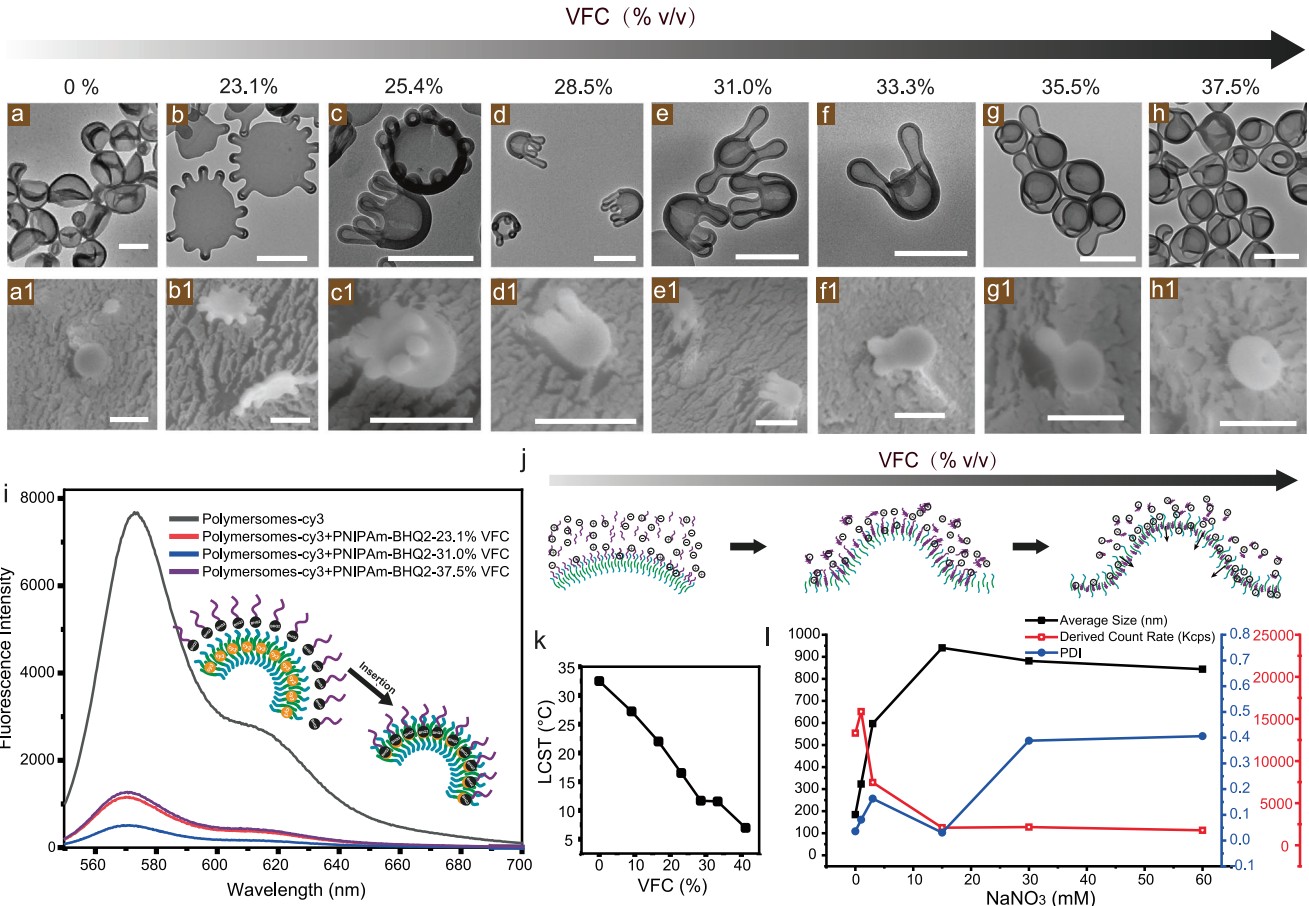

**Fig. 5 | Shape changing pathway of polymersomes with the assistance of salt ions.** TEM (**a**–**h**) and cryo-SEM (**a1**–**h1**) images of polymersomes with increase of VFC (volume fraction of co-nonsolvent) ratio from 0 to 37.5% (black arrow). **i** PNIPAm insertion followed by fluorescence spectroscopy. PS was tagged with Cy3, and PNIPAm was tagged with BHQ2. The emission of Cy3 can be quenched by BHQ2 when they are close to each other. Cy3 fluorescence emission spectrum was obtained by excitation of Cy3 at 540 nm (grey line). After 23.1% of organic solvent was added to the system, PNIPAm-BHQ2 was inserted into the polymersome membrane, and the excitation of Cy3 was quenched by BHQ2 (red line). This quenching continued both after 31%, and 37.5% of organic solvent was added to the

system (blue and purple line). **j** Schematic representation of PNIPAm-membrane interaction with the increase of VFC ratio at the presence of salt ions, salt ions assisted the insertion of PNIPAm into the membrane as single chains, an increase of organic solvent ratio facilitated a deeper insertion instead of dissociation. **k** PNIPAm LCST (lower critical solution temperature) change measured by UV-vis with the increase of VFC. **l** Dynamic light scattering measurements for the study of PNIPAm chains at 37.5% VFC with the increase of NaNO₃ concentration, the structure of PNIPAm changed from compact globule (0 mM) to swelled single chains (60 mM). PDI indicates the polydispersity index of the samples, all scale bars in the figure are 500 nm.

may need to undergo a conformational change to accommodate the PNIPAm insertion, which also requires energy during the process. When PNIPAm became hydrophobic (measured at 34 °C), a small $T\Delta S$ increase from 208 to 278 KJ/mol was observed due to the temperature increase. When NaNO₃ is added in PNIPAm solution, the anions may bind directly to the polyamide and lead to the salting-in of the polymer since NO₃⁻ is considered to be a salt-in ion. On the other hand, NO₃⁻ could also modulate the hydrophobic hydration of the hydrophobic backbone and side chains from the polymer, causing a salt-out effect. According to ITC measurements, NaNO₃ induces a slightly entropy-driven interaction with $\Delta G$ value of approximately −27.5 KJ/mol, $\Delta H$ of 0.37 KJ/mol, and $-T\Delta S$ of −27.8 KJ/mol (Supplementary Fig. 7). This suggests that the effect of NO₃⁻ on the hydrophobic main chain and hydrophobic hydration on the side chain predominates the final result (Fig. 3e), which can assist the association between the polymersome PEG corona and PNIPAm[59]. When this slightly salting out PNIPAm was added to polymersomes, the entropy increased from 208 to 397 KJ/mol, nearly twice as high as without salt (Fig. 3a). It shows that when PNIPAm is hydrophilic, ionbinding restricts the interaction between PNIPAm and PEG corona. On the other hand, salt ions might also affect the PEG corona on polymersomes. To investigate this

influence, NaNO₃ was titrated to a polymersome solution. An entropy-driven interaction was observed (Supplementary Fig. 8), suggesting that salt ions may break the hydrogen bonding between water and ether oxygen[63], which should enhance the hydrophobic association and simultaneously increase entropy (Fig. 3e). These results demonstrate that the salt-in binding of NO₃⁻ is the main reason why hydrophobic association ceases. Interestingly, when salt is titrated together with hydrophobic PNIPAm (measured 34 °C), $T\Delta S$ is around 160 KJ/mol, meaning that the heat absorbed for this hydrophobic interaction is much lower compared to when the PNIPAm is not hydrophobic (397 KJ/mol). This result indicates that NaNO₃ facilitated the hydrophobic interaction between the PEG corona and PNIPAm when forming their weak ions bindings by breaking the intrachain hydrogen bonding of polyamide, as PNIPAm might have less steric hindrance during insertion. This finding is consistent with the results from Fig. 2, which showed that when PNIPAm becomes hydrophobic due to the co-nonsolvency effect, its LCST is lowered and even cancelled (Fig. 3f). Moreover, although group 4 does not have the highest binding affinity, it provides the most binding sites compared to other groups (Fig. 3c). The lower binding affinity could be attributed to the increase in hydrophobicity. Overall, the interaction between PNIPAm and PEG

corona is an energy favoured, spontaneous process. The presence of NaNO$_3$ not only makes the interaction more accessible but also facilitates the formation of multiple arms since the interchain interaction is primarily restricted. The more collecting points can be created at the beginning of shape changing on the polymeric membrane.

## Visualization of PNIPAm insertion on polymeric membrane

To visualize PNIPAm is inserted into polymersome membrane, a negative staining method was applied before examining with TEM since PNIPAm is not visible without staining. To enhance image contrast, a heavy metal Cu$^{2+}$ was selected to coordinate with the amide group from PNIPAm and form CuO crystals before the examination (known as black solid)[27,64,65]. Here, polymersomes observed using the salt ion addition method were stained with CuSO$_4$ after removing the free PNIPAm. The presence of the copper ion complex was then examined using TEM. Black crystal dots were observed along the membrane, as shown in Fig. 4a, b, indicating that PNIPAm is inserted around the entire perimeter of these flat polymersomes with several arms. Similar to previous observation[27], PNIPAm is inserted at the edges of the pancake-like shape where the curvature is higher, providing enough space for insertion to occur. Furthermore, PNIPAm shows almost no selective gathering among the high curvature part of this shape since the amount of PNIPAm added in the solution is enough to cover the entire membrane. On the contrary, adding salt ion-assisted PNIPAm to insert individually could increase the spreading of PNIPAm. Polymersomes inserted with PNIPAm with the presence of 0, 3 and 30 mM of NaNO$_3$ were also examined with TEM (Supplementary Fig. 9), similar results were observed, PNIPAm is inserted in all the samples. Moreover, these NaNO$_3$ assisted insertion does not show a difference in the distribution of PNIPAm between the positive curvature part and negative curvature part from the membrane curvature. We assume that there might be a difference in the amount of PNIPAm inserted. However, almost no method can be used to quantify it. Further confirmation of the Cu$^{2+}$ black dots formed along the protrusions is done by TEM-energy dispersive X-ray (EDX) mapping analysis. To detect Cu signal, a Nickel (Ni) grid was used to avoid the false signal from the copper grid. Four elements (O, Cu, C, N) were mapped during the experiment. Firstly, a TEM picture of the chosen particle was taken as shown in Fig. 4c. A polymersome with multiple protrusions was stained with CuSO$_4$, resulting in the accumulation of black dots at the sites of protrusions. Subsequently, thin cross-sections of the sample were analysed, and X-ray emission of several elements were collected as shown in Supplementary Fig. 10. The spectra revealed the presence of C, N, O, Cu and Ni (from the grid). In order to visualize the distribution of each element on polymersomes, mapping analysis was applied. Carbon distribution demonstrates where the polymersome is, and the N distribution could be from the Nitrogen in PNIPAm chains. However, there might be a false signal as it is too close to C. As anticipated, Cu and O were predominantly detected at the protrusions where PNIPAm is inserted (Fig. 4c), proving that the black dots are copper ion complex and PNIPAm was primarily located at the protrusions.

Based on the above results, we have demonstrated that salt-ion-induced PNIPAm property change can improve the interaction between PNIPAm and polymersome membrane. However, the above experiments are all based on membrane flexibility, controlled by the same amount of plasticizing solvent (23.1%), which might confine the shape transformation. More morphologies can be observed when increasing the amount of plasticizer due to increased osmotic pressure and membrane flexibility[66–68]. For example, polymersomes can be changed to oblates and even closed stomatocytes without the addition of PNIPAm, through solvent addition. Moreover, the dynamic insertion of PNIPAm was observed in our previous research when salt was not added due to the PNIPAm interchain aggregation[27]. Here, similar experiments were conducted when 250 µg PNIPAm was applied in the

system with the addition of 60 mM of NaNO$_3$. The shape transformation was recorded at scheduled time points. With the addition of organic solvent (THF and dioxane), starfish-like polymersomes with multiple arms were observed at 23.1% VFC (Fig. 5b), and octopus-like shapes with multiple arms appeared at 25.4% VFC (Fig. 5c). With the increase of organic solvent ratio, the number of arms from octopus decreased to 5 (28.5%), 3/4 (31.0%), 2 (33.3%), 1 (35.5%) and 0 at 37.5% of VFC (Fig. 5d–h). The octopus half-spherical structure was examined with TEM, cryo-SEM, and cryo-TEM (Supplementary Fig. 11). Arms from octopuses are tubular structures, while the heads of are bowl-like hollow structures, arms are retracted while the opening of the stomatocyte-like structure is getting smaller and smaller. The above results indicate that polymersomes transformed to stomatocytes when the polymeric membrane is flexible enough, the same as our previous research. Moreover, neither salt ions nor PNIPAm affects the final result but only the intermediate state. The formation of octopus shape is due to the osmotic pressure from salt ions; extra osmotic pressure assisted the shape transformation of polymersomes from flat structure to bowl-like structure with the tentacles staying at the perimeter. To examine the insertion of PNIPAm with the change of solvent ratio, PS linked with Cyanine3 (Cy3) was embedded in the polymeric membrane during self-assemble. PNIPAm was tagged with a Cy3 fluorescence quencher-Black Hole Quencher 2 (BHQ2) and added to the polymersome solution together with 60 mM of NaNO$_3$. The fluorescence emitted by Cy3 was found to be quenched when PNIPAm-BHQ2 is close enough to the hydrophobic part of the membrane, as demonstrated in Fig. 5i. Their interaction was monitored by measuring the fluorescence intensity change of Cy3 after adding 23.1% (red line), 31.0% (blue line), and 37.5% (purple line) of organic solvent to the samples (Fig. 5i). Cy3 was quenched by PNIPAm-BHQ2 after 23.1% of organic solvent (THF and dioxane) addition (red line). This quenching behaviour was maintained even when polymersomes changed to stomatocytes (37.5% VFC), suggesting the insertion of PNIPAm at higher VFC. These results differ from those obtained without salt ions[27], where PNIPAm did not dissociate from the membrane, but stayed in the membrane despite its increasing flexible. On the other hand, when the solvent ratio increased from 23.1 to 37.5%, the LCST of PNIPAm dropped from 16 °C to 7 °C (Fig. 5k), indicating that PNIPAm became more hydrophobic during the experiment. Based on the above results, we propose that the PNIPAm may be deeply inserted into the membrane to prevent the membrane composition difference between the two layers from changing. To investigate the reason for the deep insertion, the behaviour of PNIPAm at a low concentration (similar to the experimental condition) was examined using the dynamic light scattering (DLS) technique. When 60 mM salt ions were present, PNIPAm did not undergo interchain aggregation at 23.1% VFC and remained separate, as few particles were detected in the solution with a large size (Fig. 2). Similar behaviour was observed when 37.5% of organic solvent was added (Fig. 5i), indicating that PNIPAm may exist as single swelled chains in the solution or mixed in the membrane. Even with the LCST of PNIPAm dropping from 16 °C to 7 °C (Fig. 5k), this interchain aggregation can be cancelled, which indicates that salt ions play a decisive role in determining the state of PNIPAm and the insertion behaviour (Fig. 5j). The above results provide us with a method of fabricating octopus-like structures. Meanwhile, the thermally responsive polymer can stay in the membrane at the arms, which indicates that stimuli-responsive shapes are developed.

To conclude, we have successfully tuned the curvature formation on the polymeric membrane by incorporating PNIPAm as a responsive hydrophobic unit and utilizing salt ions to modulate the property of PNIPAm and its interaction with the polymeric membrane. Polymersomes with multiple arms were fabricated, and the number of arms is salt concentration-dependent. The higher the salt concentration, the more arms could be formed. Moreover, the mechanism behind this salt ions assisted PNIPAm insertion was studied by investigating the salt ion

influence on PNIPAm and PNIPAm-membrane interaction. Organic solvent, water, and salt ions can form a poor solvent environment in which the PNIPAm remains partially hydrophobic. Furthermore, instead of forming interchain aggregations, PNIPAm stayed as single swelled chains, which gives them more chance to insert into the membrane. The salt ions favoured insertion was confirmed using ITC measurement since the lowest entropy was needed for a hydrophobic PNIPAm to interact with the membrane when salt ions are added. Furthermore, octopus-like shapes were observed when membranes became more flexible with PNIPAm that was embedded in the membrane for potential stimuli-responsive nanomotors. As salt ions showed a similar influence on PNIPAm and proteins[30–33], we hope this study can provide evidence for understanding the role of salt ions in biomembrane curvature formation. This first strategy also allows us to fabricate polymer-based nano-systems with stimuli-responsive membranes for various applications.

## Methods

### Fabrication of polymersomes
10 mg $PEG_{44}$-b-$PS_n$ was dissolved in 1 mL organic solvent mixture of tetrahydrofuran (THF):1,4-dioxane (4:1) by volume. A stirring plate was used to dissolve the polymer for 30 min with a stirring bar inside the solution. 3 mL of Milli-Q water was then dropped into the polymer solution at a rate of 1 mL h$^{-1}$ under vigorous stirring (900 rpm). Polymers self-assembled to polymersomes during water addition and this polymersome suspension was then transferred to dialysis membrane (SpectraPor, molecular weight cut-off: 12,000–14,000) and dialysed against pure water (1000 mL) to remove the organic solvent for 48 h with a frequent change of water.

### PNIPAm-membrane interaction with solvent addition method
Firstly, 465 µL of rigid polymersome solution (2 mg/mL) was transferred to a 5 mL vial, 5 µL of PNIPAm with 50 mg/mL concentration and 30 µL of NaNO$_3$ solutions with various concentrations (0 M, 0.0167 M 0.05 M, 0.25 M, 0.5 M, 1 M) were then added into the polymersome solutions, respectively. THF:dioxane (4:1 v/v) mixture was added at a rate of 300 µL h$^{-1}$ via syringe pump under a stirring speed of 900 rpm after 10 min of mixing. Samples were withdrawn and quenched as scheduled, and examined using TEM, Cryo-TEM and Cryo-SEM.

### PNIPAm transmittance measurements using UV-vis
The transmittance of PNIPAm samples were measured on a JASCO V-630 UV-Vis spectrophotometer equipped with a thermoregulator (±0.1 °C) with deionized water as a reference (100% transmittance) at 500 nm wavelength. 60 µL NaNO$_3$ solutions with various concentrations (0 M, 0.0167 M 0.05 M, 0.25 M, 0.5 M, 1 M) were added to 940 µL PNIPAm solutions at different water/organic solvent mixtures. PNIPAm concentration was kept at 2 mg/mL. The LCST values were determined at 50% of transmittance.

### Shape transformation of polymersomes with the increase of organic solvent ratio
465 µL of rigid polymersomes solution was transferred into a 5 mL vial followed with 250 µg of PNIPAm and 60 mM of NaNO$_3$ then added into the polymersome solutions. After 10 min of mixing, THF:dioxane (4:1 v/v) mixture was added via syringe pump under a stirring speed of 900 rpm at the rate of 300 µL h$^{-1}$. Samples were taken at different time intervals (30, 35, 40, 45, 50, 55, 60 min), quenched and examined via TEM and Cryo-SEM.

### Size examination of PNIPAm aggregates formation
Dynamic light scattering (DLS) experiments were conducted on a Malvern Zetasizer Nano S at 25 °C. A 0.22-µm filter was used to remove the possible dust in PNIPAm solutions with different amount of NaNO$_3$

from 0 mM to 60 mM added. Pure THF and 1,4-dioxane (4:1) was then added dropwise to the solution and mixed properly before measurement. The final concentration of PNIPAm in different water/organic solvent mixtures was kept at 0.1 mg/mL.

## Data availability
All data generated and analysed during this study are included in this article and its Supplementary Information, and also available from the authors upon request.

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

## Acknowledgements

D.A. Wilson acknowledges the NWO Chemiche Wetenschappen VIDI Grant 723.015.001 for financial support. We acknowledge support from the Ministry of Education, Culture and Science (Gravity Program 024.001.035) and the European Union's Horizon 2020 research and innovation programme under the Marie Sklodowska-Curie grant agreement No. 891484. The authors would like to thank G. J. Janssen from General Instruments at Radboud University for assistance with the microscopy analysis.

## Author contributions

J.S. and D.A.W. designed the experiments. J.S., J.L., W.L., S.Z., D.Wa. and G.U. performed the experiments, J.S. analysed the results, S.K. performed the simulation of the stomatcocyte polymersomes. J.S. and D.A.W. wrote the manuscript.

## Competing interests

The authors declare no competing interests.
