## [Peer Review File · Nature Communications]

Morphogenesis of Starfish PolymersomesREVIEWER COMMENTS

Reviewer #1 (Remarks to the Author):

This paper continues authors' previous work (Ref. 27) on the shape change of spherical polymersomes by implementing PNIPAM to the hydrophobic domain of the polymersome membrane consisting of PEG-b-PS. The driving force for the shape change of polymersomes was suggested to be osmotic pressure caused by the addition of organic solvents, which also rendered PNIPAM more hydrophobic as indicated by the reduced LCST. The implementation of PNIPAM in the polymersome membrane alters the membrane properties (bending resistivity or membrane elasticity), causing the formation of starfish-like vesicles upon the reduction of the internal volume of the hydrophilic compartment of polymersomes. In this work, NaNO₃ was used to provide (1) additional osmotic pressure and (2) interaction with PNIPAM rendering the thermosensitive polymer more hydrophobic by exerting 'salting-out' effect. The shape change of polymersomes was correlated to the concentration of NaNO₃, which increases the number of protrusions at the surface of polymersomes. Of particular interest is the ITC measurements to show the thermodynamically favorable interactions between PNIPAM and (or) PEG with NaNO₃. (with this experiment, one particular question is how the authors confirmed the concentration of polymeric vesicle solutions, 2.5 mM? is this the concentration of the block copolymer used or the actual vesicle concentration? For the first case, is this experiment valid? or for the second case, is the concentration accurate?). The effect of the amount of organic solvents on the shape change was studied. And, finally, the fluorescence quenching experiment with Cy3-modified PS-embedded polymersomes with BHQ-modified PNIPAM was conducted to show that the fluorescence was quenched by the implementation of PNIPAM to the membrane in the presence of organic solvents. This work merits as a significant advance of the previously reported experiments, but fails to deliver enough new insights to warrant the publication in Nature Communications. Additional experiments with other salts in the Hoffmeister series could clear the point that the presence of NaNO₃ is not only for imposing additional osmotic pressure, but also introduces specific interaction between PNIPAM/PEG and ions.

Reviewer #2 (Remarks to the Author):

In the present paper, Sun et al. discuss the addition of poly(N-isopropylamide) (PNIPAm) as a means to control the shape change in polystyrene-poly(ethylene glycol) block copolymers. Namely, they look at the combination of PNIPAm with organic solvent, water and salt ions. They investigate the thermodynamics of the assembly by isothermal titration calorimetry.

The paper proposes a fundamentally interesting concept for the idea of shape change. However, this reviewer believes that many the great results obtained in this manuscript are somewhat overshadowed by the presentation of said results. The absence of chemical structures, the fact that acronyms not defined (e.g. PNIPAm and ITC in the abstract; PEG in the introduction), the absence of chemical structures and the dearth of figures make it very hard to follow even for people working in the field. In addition, the writing is not particularly concise and to-the-point, with lots of run-on sentences (see

below) which detract from the results.

Specific comments:

In the introduction, “researchers was able to” should be replaced by “were able to” and there should be a comma after “stomatocyte-in-stomatocyte”.

At the end of the first page, the opening clause “To mimic the protein interaction induced shape transformation in polymersomes” does not make sense. This reviewer suspects that there is a compound adjective somewhere. Do the authors mean protein-induced shape transformation? What interaction would cause the transformation?

The authors finally describe poly(N-isopropylamide) on page 3, it should not be capitalized and the N should be italicized.

Page 4, “theextra osmotic pressure added” should be revised to “the extra osmotic pressure experienced by”. Saying that osmotic pressure is added seems strange.

In the following sentence, this reviewer assumes that the authors mean “salt-ion-induced reduction”. Otherwise, the sentence does not make sense. Generally, throughout the authors should pay attention to compound adjective rules (e.g. entropy-driven, salt-in, salt-out, etc.)

The sentence starting with “Without salt ions” a couple lines below is a run on and is as such unintelligible. Why would polymersomes be transferred to flat disks? Do the authors mean that they will be transformed? The stomatocytes are half-opened not “half open”.

The TEM justification for the parametrization is not very strong. The authors show one image. The supporting information does not indicate an extensive analysis of the different polymersomes either. To be believable, the authors would need to perform said analysis on many micrographs and many assembled structures.

Page 5, osmotic pressure cannot be given. Consider rephrasing.

The sentence starting with “Volume reduction has been considered” is a run-on sentence. Consider splitting it into two. The second fragment as written does not make sense.

The following sentence needs a reference to the previous work of the authors. The reader cannot be expected to know off the top of their heads the body of work of the authors.

A reference to the change in hydrophobicity of PNIPAm is needed. Also, the authors discuss the addition of NaNO₃ and then follow with the sentence: “This is based on the fact that PNIPAm can change its property from hydrophilic to hydrophobic when certain organic solvent is added.” It is unclear how the addition of an inorganic compound is related to that. There is no apparent correlation at this point. The authors should indicate clearly what organic solvent they are adding.

While this reviewer also tends to believe that these starfish structures are flat with protruding arms nothing prevents them from looking more like a nitrile glove that was inflated. Do the authors have additional proof of the flat nature of the center portion? Also, the authors should comment on the tip of the arms; Figure 1 m1 and n1 have especially interesting features.

The authors discuss kosmotropes and chaotropes but this reviewer could not identify any experiments that demonstrated this effect here beyond the use of NaNO₃. Is that an oversight? Is that a planned experiment? If so, it should appear in the conclusion rather than here.

As mentioned above with the organic solvent, the authors should define the co-nonsolvent used. Is that THF and dioxane? It is neither clearly mentioned in the text nor in the supplementary information.

The sentence starting with “As a weak chaotrope” is a run-on. Consider splitting it into two. There are

multiple instances where the authors start with “[long clause 1], however, [long clause 2]”. Most of them can be split. Also, consider a different adverb to express contradiction than however to make it less monotonous as it is repeated many times (e.g. nevertheless, yet, although, whereas, energy-favoured...)

The sentence indicating that “hydrophobic hydration was destroyed” will be puzzling to the non-specialist. The authors should define it, especially, as it is used later in the document

Above Figure 2, the authors discuss “the size and light scattering intensity”. The authors need to explain what size they are referring to.

Right below, the sentence starting with “However” is a run-on and is unintelligible.

Figure 2, the caption title should also capture the variation in VFC. Figure 2b is presumably extracted from Figure 2c. This reviewer does not really understand the advantage of showing Figure 2b.

Figure 2d is not particularly clear. It is too small and does not really convey whatever the message the authors intend to pass onto the reader.

While this reviewer appreciates the limitations of ITC, the assumption that temperature increase in PNIPAm is equivalent to solvent addition is bold. Is there some literature precedent that the authors can point to? Is there any calibration or experiment that can be done to demonstrate the veracity of this assertion?

Page 14, hydrophobic instead of hydrophob

Generally speaking, the explanation of the ITC results is both not thorough in terms of describing what is observed and what it means. This section needs to be improved upon for instance by providing actual number rather than qualitative explanation. The section also reads very much like a sequence of suppositions and the reader is left with the impression that the results are not fully understood. What do the authors mean by “flat but armed”. “The same as previous observation” should be rephrased.

Page 16, “it’s too close” should be “it is too close”.

The demonstration about insertion via quenching should appear way earlier in the text. It is a bit out of place at the end.

Overall, this work constitutes a highly worthy piece of work that should definitely be published. This reviewer however questions the novelty, especially, in light of a paper based on a similar platform by the authors in Nat. Comm. earlier this year. In addition, there are some minor technical questions that need to be addressed along with more important (yet manageable) writing issues.

REVIEWER COMMENTS

Reviewer #1 (Remarks to the Author):

This paper continues authors' previous work (Ref. 27) on the shape change of spherical polymersomes by implementing PNIPAm to the hydrophobic domain of the polymersome membrane consisting of PEG-b-PS. The driving force for the shape change of polymersomes was suggested to be osmotic pressure caused by the addition of organic solvents, which also rendered PNIPAM more hydrophobic as indicated by the reduced LCST. The implementation of PNIPAM in the polymersome membrane alters the membrane properties (bending resistivity or membrane elasticity), causing the formation of starfish-like vesicles upon the reduction of the internal volume of the hydrophilic compartment of polymersomes. In this work, NaNO₃ was used to provide (1) additional osmotic pressure and (2) interaction with PNIPAM rendering the thermosensitive polymer more hydrophobic by exerting 'salting-out' effect. The shape change of polymersomes was correlated to the concentration of NaNO₃, which increases the number of protrusions at the surface of polymersomes. Of particular interest is the ITC measurements to show the thermodynamically favorable interactions between PNIPAm and (or) PEG with NaNO₃. (with this experiment, one particular question is how the authors confirmed the concentration of polymeric vesicle solutions, 2.5 mM? is this the concentration of the block copolymer used or the actual vesicle concentration? For the first case, is this experiment valid? or for the second case, is the concentration accurate?)

Answer: We thank the reviewer for the comments. The concentration of the polymeric vesicle solution we have mentioned in the ITC experiment is based on the block copolymer (PEG-b-PS) concentration we used. In principle, this concentration should be accurate, and the reason why we choose to use this concentration is to have a direct comparison between PNIPAm concentration and PEG concentration, as the interaction is supposed to happen between these two, we were wondering whether there is a correlation between these two, but after several tries, we cannot really make a conclusion or discussion on this. Moreover, the experiment is for sure valid since exploring the correlation PNIPAm and particle concentration is not the primary purpose. Thanks very much for pointing out the problem. We do agree that we need to clarify this concentration. Therefore, we have made some changes, accordingly, as shown below:

The experiments were conducted by titrating PNIPAm into a polymeric vesicle solution assembled from 2.5 mM PEG-b-PS as shown in Fig. 3a-b. 4 groups of heat exchange were measured, including the interaction between PNIPAm and polymeric vesicles when PNIPAm were hydrophilic (25 °C) or hydrophobic (34 °C) and when salt ions were added into the above two groups. The dilution of PNIPAm was set as a control group during the measurement.

The effect of the amount of organic solvents on the shape change was studied. And, finally, the fluorescence quenching experiment with Cy3-modified PS-embedded polymersomes with BHQ-modified PNIPAM was conducted to show that the fluorescence was quenched by the implementation of PNIPAm to the membrane in the presence of organic solvents. This work merits as a significant advance of the previously reported experiments, but fails to deliver enough new insights to warrant the publication in Nature Communications.

Answer: We thank the reviewer for the comments. We have recently published a paper in Nature Communications with the PNIPAm addition method. However, we believe that does not decrease the novelty of this manuscript since the previous manuscript had a different focus while here we investigate for the first time the observance of morphologies such as the octopus-like shapes and their mechanism of formation, which researchers from various fields will appreciate. Although PNIPAm based method has been published, we believe the novelty of this manuscript is its development of new and valuable morphologies with feasible and easy method and our focus on its mechanism of formation. The coronavirus challenge brought scientists together to develop nanocarriers for vaccine development with unprecedented speed. More and more attention has been placed on efficient designs of delivery vehicles with controlled shapes, structures, and functions. Polymeric vesicles have become promising tools in the fields of micro/nanoreactors, drug delivery, and cell mimicking due to their enhanced membrane stiffness and chemical versatility in terms of flexibility and chemical design of the membrane and its properties. However, the scant shapes obtained from polymersomes restricted their full potential. For example, in nanomedicine, the shape of nanocarriers is considered a new parameter in governing their performance in vivo, as important as polymer property, particle size and surface chemistry. Therefore, shape control methods hold the promise to generate polymersomes with different morphologies and perform shape transformations on the bilayer membranes, which are not yet available nevertheless expected from scientists in different fields.

In this manuscript, we have designed a new approach for the formation of multi-armed polymersomes by applying PNIPAm as a responsive hydrophobic unit and salt ions to modulate the property of PNIPAm and its interaction with the polymeric membrane. The use of salt ions is innovative and feasible. Polymersomes with multiple arms were fabricated, and the number of arms is salt concentration-dependent. Moreover, the mechanism behind this salt ions assisted PNIPAm insertion was studied by investigating the salt ion influence on PNIPAm and on PNIPAm-membrane interaction. This is the first time that the multi-armed polymersomes have been formed and studied, and also the first-time detailed information of curvature generation has been revealed in a membrane model. In the future, we hope this study can provide evidence for understanding the role that salt ions play in biomembrane curvature formation. This new strategy also allows us to fabricate polymer-based nano-

systems with stimuli-responsive membranes for various applications. Our focus on the mechanism study about the salt influence in shape changing could provide evidence to support the role of salt ions in the formation of curvature in polymeric membranes and understand natural processes where salt ions and membrane proteins play crucial roles. Our knowledge of this field suggests that the importance of this timely work will be appreciated by researchers from diverse backgrounds, and we also want to promote this work and hope more and more researchers from different fields can see this work and make it worthwhile.

Additional experiments with other salts in the Hofmeister series could clear the point that the presence of NaNO_3 is not only for imposing additional osmotic pressure, but also introduces specific interaction between PNIPAM/PEG and ions.

Answer: We thank the reviewer for the comment. The relevant experiments have been conducted and discussed in the manuscript. As shown below:

As shown in Supplementary Fig 5, we conducted control experiments to further explore ion influence on membrane curvature formation with both strong Hofmeister “kosmotropes” and “chaotropes”. Interestingly, strong “chaotropes” ions like NaSCN showed a positive influence on the construction of membrane curvature with an average of 11 arms formed, which is even slightly higher than NaNO_3 . Moreover, NaCl , with a similar Hofmeister effect as NaNO_3 , showed a similar impact on membrane curvature formation. On the other hand, when “kosmotropes” ions are added in the solution, the formation of membrane curvature is not as much noticeable as when NaNO_3 was added. Two “kosmotropes” ions Na_2HPO_4 and Na_2SO_4 , were added separately in these experiments, and both of them were found less effective than the “chaotropes”. Na_2SO_4 showed a much less influence in the formation of curvatures, 3-5 arms were formed on average. In general, “chaotropes” were found to penetrate the headgroup region of a hydrocarbon-packed monolayer and even lipid membranes since the presence of chaotropic salts in the subphase increased the surface pressure^{45,46}. We believe that the chaotropic salts in our experiment had a similar effect on the polymeric membrane.

Supplementary Figure 5. Shape transformation of polymersomes with Hofmeister salt. 60 mM NaSCN, NaCl, NaNO₃, Na₂HPO₄, and Na₂SO₄ were added to polymersomes together with 250 μg of PNIPAm to explore the salt impact on membrane curvature formation. After 23% of organic solvent was added to the systems, samples were quenched and examined by TEM. Scale bar 1 μm.

Reviewer #2 (Remarks to the Author):

In the present paper, Sun et al. discuss the addition of poly(N-isopropylamide) (PNIPAm) as a means to control the shape change in polystyrene-poly(ethylene glycol) block copolymers. Namely, they look at the combination of PNIPAm with organic solvent, water and salt ions. They investigate the thermodynamics of the assembly by isothermal titration calorimetry.

The paper proposes a fundamentally interesting concept for the idea of shape change. However, this reviewer believes that many the great results obtained in this manuscript are somewhat overshadowed by the presentation of said results. The absence of chemical structures, the fact that acronyms not defined (e.g. PNIPAm and ITC in the abstract; PEG in the introduction), the absence of chemical structures and the dearth of figures make it very hard to follow even for people working in the field. In addition, the writing is not particularly concise and to-the-point, with lots of run-on sentences (see below) which detract from the results.

Answer: We thank the reviewer for the comment. We have defined the acronyms and added them in the manuscript. The chemical structures and their interactions are also described now in Figure3.

Specific comments:

In the introduction, "researchers was able to" should be replaced by "were able to" and there should be a comma after "stomatocyte-in-stomatocyte".

Answer: We thank the reviewer for the comment. The sentences have been corrected.

At the end of the first page, the opening clause "To mimic the protein interaction induced shape transformation in polymersomes" does not make sense. This reviewer suspects that there is a compound adjective somewhere. Do the authors mean protein-induced shape transformation? What interaction would cause the transformation?

Answer: We thank the reviewer for the comment. We indicate that some membrane proteins can insert into the lipid membrane and cause the shape transformation of cells. We have rephrased this sentence. As shown below:

To mimic the protein-membrane interaction induced shape transformation in cells, a new strategy has been developed recently in our group, in which poly(*N*-isopropylacrylamide) (PNIPAm) was used as alternatives for proteins with hydrophobic unit to insert into the polymersome membrane²⁷.

The authors finally describe poly(*N*-isopropylamide) on page 3, it should not be capitalized and the *N* should be italicized.

Answer: We thank the reviewer for the comment. The mistake has been rectified.

Page 4, "theextra osmotic pressure added" should be revised to "the extra osmotic pressure experienced by". Saying that osmotic pressure is added seems strange.

Answer: We thank the reviewer for the comment. This sentence has been rephrased.

In the following sentence, this reviewer assumes that the authors mean "salt-ion-induced reduction". Otherwise, the sentence does not make sense. Generally, throughout the authors should pay attention to compound adjective rules (e.g. entropy-driven, salt-in, salt-out, etc.)

Answer: We thank the reviewer for the comments. The mistakes have been rectified.

The sentence starting with "Without salt ions" a couple lines below is a run on and is as such unintelligible. Why would polymersomes be transferred to flat disks? Do the authors mean that they will be transformed? The stomatocytes are half-opened not "half open".

Answer: We thank the reviewer for the comments. Here we mean “transformed”, the polymersomes can be transformed to flat disks due to solvent exchange during the process. The sentence has been rephrased as shown below:

Without salt ions, polymersomes can be transformed to flat disks (Fig.1a1-a2, g). However, the addition of salt ions pushed the shape transformation further to half-open stomatocytes-like polymersomes (Fig. 1h) due to the extra inner volume reduction. It can be seen by the reduced distance between two bilayers from the EM images.

The TEM justification for the parametrization is not very strong. The authors show one image. The supporting information does not indicate an extensive analysis of the different polymersomes either. To be believable, the authors would need to perform said analysis on many micrographs and many assembled structures.

Answer: We thank the reviewer for the comments. We apologize for the unclear description. The fitted phase diagram is based on 5 assembled structures at each experiment condition. We did not perform a much bigger analysis since it is not that easy to collect a larger sample size in a cryo-TEM experiment and the volume reduction is very clear between groups.

Page 5, osmotic pressure cannot be given. Consider rephrasing.

The sentence starting with "Volume reduction has been considered" is a run-on sentence. Consider splitting it into two. The second fragment as written does not make sense.

The following sentence needs a reference to the previous work of the authors. The reader cannot be expected to know off the top of their heads the body of work of the authors.

Answer: We thank the reviewer for the comments. The sentences have been rephrased and the reference has been added as shown below:

When 1 mM, 3 mM, 15 mM and 60 mM NaNO₃ were added in the system, Δv decreased 49%, 62%, 64% and 66% respectively comparing to 0 mM salt (Fig. 1f). It indicates that adding salt ions can decrease the inner volume of polymersomes since the polymersome membrane has experienced extra osmotic pressure³⁷. Besides, the volume reduction becomes more difficult with the inner volume getting smaller since bending energy is higher at the membrane. Decreasing inner volume resulted in almost no difference in shape transformation. Although volume reduction has been considered the driving force in the formation of structures like multi-armed starfish *etc* in liposome systems^{34,35,38}, the above results proved that tuning only the volume reduction will not lead to a further shape transformation. As studied in our previous work²⁷, membrane composition change was considered the

critical condition for shape changing. We, therefore, assume that membrane composition change could be the main factor in the formation of arms.

A reference to the change in hydrophobicity of PNIPAm is needed. Also, the authors discuss the addition of NaNO₃ and then follow with the sentence: "This is based on the fact that PNIPAm can change its property from hydrophilic to hydrophobic when certain organic solvent is added." It is unclear how the addition of an inorganic compound is related to that. There is no apparent correlation at this point. The authors should indicate clearly what organic solvent they are adding.

Answer: We thank the reviewer for the comments. The sentences have been rephrased and the reference has been added as shown below:

As studied in our previous work²⁷, membrane composition change was considered the critical condition for shape changing. We, therefore, assume that membrane composition change could be the main factor in the formation of arms. To prove this hypothesis, 250 µg PNIPAm was added to the solution together with different amounts of NaNO₃. After 23% organic solvent was added to the solutions, samples were quenched and examined with TEM (Fig1.i1-n1) and Cryo-SEM (Fig1.i2-n2). PNIPAm can change its behaviour from hydrophilic to hydrophobic when a certain amount of organic solvent (THF, Methanol, etc.) is added to its solution³⁹. We therefore expected PNIPAm to insert itself into polymeric membranes as a hydrophobic motif which leads to further changing of the membrane composition based on hydrophobic effect²⁷. With the increase of salt concentration, polymersomes changed from boomerang-like shapes, tubes, and three armed shapes to shapes with multiple arms. The higher the salt concentration, the more arms formed on polymersomes. When salt concentration reached 15 mM, 6-7 long arms were observed. When salt concentration reached 30 mM, 7-8 short arms were observed, and the number of arms could be above 10 when 60 mM salt was added.

While this reviewer also tends to believe that these starfish structures are flat with protruding arms nothing prevents them from looking more like a nitrile glove that was inflated. Do the authors have additional proof of the flat nature of the center portion? Also, the authors should comment on the tip of the arms; Figure 1 m1 and n1 have especially interesting features.

Answer: We thank the reviewer for the comments. The inflated nitrile glove-like structures can also be achieved under different conditions. This manuscript will not discuss these results as the experimental conditions under which are obtained are different. Figure1m1 and n1 are TEM pictures; therefore, there is some drying effect on the structures. That is why it is necessary to check the structures with cryo-SEM. With the cryo-SEM picture, more details can be observed. The flat nature

of the center portion can be proved from cryo-SEM pictures taken from different angles, as shown below.

The authors discuss kosmotropes and chaotropes but this reviewer could not identify any experiments that demonstrated this effect here beyond the use of NaNO_3 . Is that an oversight? Is that a planned experiment? If so, it should appear in the conclusion rather than here.

Answer: We thank the reviewer for the comments. Additional experiments have been performed to complete this discussion as shown below:

As shown in Supplementary Fig 5, we conducted control experiments to further explore ion influence on membrane curvature formation with both the strong Hofmeister “kosmotropes” and “chaotropes”. Interestingly, strong “chaotropes” ions like NaSCN showed a positive influence on the construction of membrane curvature with an average of 11 arms formed, which is even slightly higher than NaNO_3 . Moreover, NaCl , with a similar Hofmeister effect as NaNO_3 , showed a similar impact on membrane curvature formation. On the other hand, when “kosmotropes” ions are added in the solution, the formation of membrane curvature is not as much as when NaNO_3 was added. Two “kosmotropes” ions Na_2HPO_4 and Na_2SO_4 , were added separately in these experiments, and both of them were found less effective than the “chaotropes”. Na_2SO_4 showed a much less influence in the formation of curvatures, 3-5 arms were formed on average. In general, “chaotropes” were found to penetrate the headgroup region of a hydrocarbon-packed monolayer and even lipid membranes since the presence of chaotropic salts in the subphase increased the surface pressure^{44,45}. We believe the chaotropic salts in our experiment had a similar effect on the polymeric membrane.

Supplementary Figure 5. Shape transformation of polymersomes with Hofmeister salt. 60 mM NaSCN, NaCl, NaNO₃, Na₂HPO₄ and Na₂SO₄ were added to polymersomes together with 250 μg of PNIPAm to explore the salt impact on membrane curvature formation. After 23% of organic solvent was added to the systems, samples were quenched and examined by TEM. Scale bar 1 μm.

As mentioned above with the organic solvent, the authors should define the co-nonsolvent used. Is that THF and dioxane? It is neither clearly mentioned in the text nor in the supplementary information. The sentence starting with "As a weak chaotrope" is a run-on. Consider splitting it into two. There are multiple instances where the authors start with "[long clause 1], however, [long clause 2]". Most of them can be split. Also, consider a different adverb to express contradiction than however to make it less monotonous as it is repeated many times (e.g. nevertheless, yet, although, whereas, energy-favoured...)

Answer: We thank the reviewer for the comments. The organic solvent mixture that is mentioned in this manuscript refers to THF and dioxane. We have made it clear in the manuscript, and the sentences have been rephrased accordingly.

The sentence indicating that "hydrophobic hydration was destroyed" will be puzzling to the non-specialist. The authors should define it, especially, as it is used later in the document.

Answer: We thank the reviewer for the comments. The hydrophobic hydration has been defined in the manuscript.

Above Figure 2, the authors discuss "the size and light scattering intensity". The authors need to explain what size they are referring to.

Answer: We thank the reviewer for the comments. We have made it clear in the manuscript, as shown below:

Furthermore, when the salt concentration increased to 30 and 60 mM, the size and light scattering intensity of these samples examined with DLS did not significantly change, meaning that the particles were not formed. Although the UV-Vis experiments demonstrated the hydrophobicity of PNIPAm, DLS results revealed that PNIPAm chains are still separated into single chains when they turn hydrophobic and do not self-assemble into particles due to the salt ions that bind onto the polymer chain during the process (Fig. 2d). This instead facilitates the insertion of PNIPAm into the polymeric membrane.

Right below, the sentence starting with "However" is a run-on and is unintelligible.

Answer: We thank the reviewer for the comment. The sentences have been rephrased.

Figure 2, the caption title should also capture the variation in VFC. Figure 2b is presumably extracted from Figure 2c. This reviewer does not really understand the advantage of showing Figure 2b.

Figure 2d is not particularly clear. It is too small and does not really convey whatever the message the authors intend to pass onto the reader.

Answer: We thank the reviewer for the comments. We agree with the reviewer's suggestion and have made some changes to this figure. In addition, we made Figure 2e clear, and the meaning of each symbol has also been clarified as shown below:

Figure 2. The co-nonsolvency phenomenon of PNIPAm at different salt concentrations with VFC variate from 0% to 41%. (a) PNIPAm transmittance measurement by UV-vis spectroscopy to detect the shift of LCST with different salt concentrations (0 mM- 60 mM) at 23% of VFC. (b) The shift of LCST with VFC varies from 0% to 41%. (c) Dynamic light scattering measurements to observe the hydrophobic behaviour of PNIPAm at the VFC of 23% with the addition of 0, 1, 3, 15, 30, 60 mM NaNO₃ in PNIPAm water solution. (d) Chemical structure of PNIPAm and possible salt interaction position between PNIPAm and salt ions. (e) Schematic representation of hydrophobic behaviour of PNIPAm chains at the addition of salt ions and solvent, salt-only cannot turn PNIPAm to hydrophobic, but after addition of organic solvent (THF and dioxane), PNIPAm became hydrophobic, however, no interchain aggregation was observed.

While this reviewer appreciates the limitations of ITC, the assumption that temperature increase in PNIPAm is equivalent to solvent addition is bold. Is there some literature precedent that the authors can point to? Is there any calibration or experiment that can be done to demonstrate the veracity of this assertion?

Answer: We thank the reviewer for the constructive comments. As mentioned, the ITC has restrictions on the use of organic solvents. Therefore, we tried to think of a better way to explain our finding, and temperature increase is used as an alternative strategy in our experiment. Its theoretical basis is that

lower critical solution temperature (LCST) of PNIPAm in water is believed to be strongly related to the destabilization of hydrogen bonds between water molecules and amide groups with increasing temperature, induced by the presence of the hydrophobic isopropyl group and backbone^{1,2}. The mixing process at low temperatures is favored by the formation of hydrogen bonds thermodynamically, which leads to a large negative enthalpy of mixing³. When a mixed solvent is added to the solution, PNIPAm solubility is reduced within a range of intermediate solvent concentrations in binary aqueous solutions. This phenomenon suggested that 'water-solvent complexes' are preferred to PNIPAm-water hydrogen bonds⁴. This indicates that the LCST changes share similar chemistry. Moreover, systematic research has been done to compare these changes⁵. Using IR and micro-Raman spectroscopy, the authors showed that gradual redshifts of the C-H stretching, and the amide II bands happen with increasing temperature or DMSO concentration. Therefore, we believe our experiment result can help us and the readers to understand these interactions behind the shape changes. We have added this reference in our manuscript to make the discussion more solid.

Page 14, hydrophobic instead of hydrophob/

Answer: We thank the reviewer for the comment. The mistake has been rectified.

Generally speaking, the explanation of the ITC results is both not thorough in terms of describing what is observed and what it means. This section needs to be improved upon for instance by providing actual number rather than qualitative explanation. The section also reads very much like a sequence of suppositions and the reader is left with the impression that the results are not fully understood.

Answer: We thank the reviewer for the constructive comments. We have revised this section of the manuscript. We hope the revised discussion is clear and easy to understand.

What do the authors mean by "flat but armed". "The same as previous observation" should be rephrased.

Answer: We thank the reviewer for the comments. Here we mean that these polymersomes have several arms, and at the same time, they are still flat disk structures. We corrected the sentence to make it more clear, as shown below:

Black crystal dots were observed along the membrane, as shown in Fig. 4a-b, meaning that PNIPAm is inserted all over the perimeter of these flat polymersomes with several arms. Similar to our previous observation²⁷, PNIPAm is inserted in the edge of the pancake-like shape where the curvature is higher so that there is enough space for the insertion to occur.

Page 16, "it's too close" should be "it is too close".

Answer: We thank the reviewer for the comments. We revised this sentence.

The demonstration about insertion via quenching should appear way earlier in the text. It is a bit out of place at the end.

Answer: We thank the reviewer for the comments. The insertion demonstration is also shown in Figure 3f. Here we mainly want to demonstrate the insertion with different amounts of organic solvent added during the shape transformation.

Overall, this work constitutes a highly worthy piece of work that should definitely be published. This reviewer however questions the novelty, especially, in light of a paper based on a similar platform by the authors in Nat. Comm. earlier this year. In addition, there are some minor technical questions that need to be addressed along with more important (yet manageable) writing issues.

Answer: We thank the reviewer for the comments. We have fully listened to your comments and have comprehensively revised this manuscript. Some other experiments have also been added accordingly. We hope that the revised article will meet your requirements, and further suggestions are appreciated.

- 1 Schild, H. G. & Tirrell, D. A. Microcalorimetric detection of lower critical solution temperatures in aqueous polymer solutions. *The Journal of Physical Chemistry* **94**, 4352-4356 (1990).
- 2 Ilmain, F., Tanaka, T. & Kokufuta, E. Volume transition in a gel driven by hydrogen bonding. *Nature* **349**, 400-401 (1991).
- 3 Walker, J. S. & Vause, C. A. Reappearing Phases. *Scientific American* **256**, 98-105 (1987).
- 4 Costa, R. O. R. & Freitas, R. F. S. Phase behavior of poly(N-isopropylacrylamide) in binary aqueous solutions. *Polymer* **43**, 5879-5885 (2002).
- 5 Yamauchi, H. & Maeda, Y. LCST and UCST Behavior of Poly(N-isopropylacrylamide) in DMSO/Water Mixed Solvents Studied by IR and Micro-Raman Spectroscopy. *The Journal of Physical Chemistry B* **111**, 12964-12968 (2007).

Reviewers' comments:

No comments were made to the authors only comments to the editor.

REVIEWER COMMENTS

Reviewer #1 (Remarks to the Author):

This paper continues authors' previous work (Ref. 27) on the shape change of spherical polymersomes by implementing PNIPAm to the hydrophobic domain of the polymersome membrane consisting of PEG-b-PS. The driving force for the shape change of polymersomes was suggested to be osmotic pressure caused by the addition of organic solvents, which also rendered PNIPAM more hydrophobic as indicated by the reduced LCST. The implementation of PNIPAM in the polymersome membrane alters the membrane properties (bending resistivity or membrane elasticity), causing the formation of starfish-like vesicles upon the reduction of the internal volume of the hydrophilic compartment of polymersomes. In this work, NaNO₃ was used to provide (1) additional osmotic pressure and (2) interaction with PNIPAM rendering the thermosensitive polymer more hydrophobic by exerting 'salting-out' effect. The shape change of polymersomes was correlated to the concentration of NaNO₃, which increases the number of protrusions at the surface of polymersomes. Of particular interest is the ITC measurements to show the thermodynamically favorable interactions between PNIPAm and (or) PEG with NaNO₃ (with this experiment, one particular question is how the authors confirmed the concentration of polymeric vesicle solutions, 2.5 mM? is this the concentration of the block copolymer used or the actual vesicle concentration? For the first case, is this experiment valid? or for the second case, is the concentration accurate?)

Answer: We thank the reviewer for the comments and suggestions. We do agree that we need to clarify this concentration. The concentration of the polymeric vesicle solution we have mentioned in the ITC experiment is based on the block copolymer (PEG-b-PS) concentration we used. In principle, this concentration should be accurate, and the reason why we choose to use this concentration is to have a direct comparison between PNIPAm concentration and PEG concentration, as the interaction expected to occur between these two, as we were wondering whether there is a correlation between these two. Nevertheless, after several trials, we could not draw clear conclusions or discussion on this. Moreover, the experiment is for sure valid since exploring the correlation PNIPAm and particle concentration is not the primary focus. We have made some changes, accordingly, as shown below to clarify this:

The experiments were conducted by titrating PNIPAm into a polymeric vesicle solution assembled from 2.5 mM PEG-b-PS as shown in Fig. 3a-b. 4 groups of heat exchange were measured, including the interaction between PNIPAm and polymeric vesicles when PNIPAm were hydrophilic (25 °C) or hydrophobic (34 °C) and when salt ions were added into the above two groups. The dilution of PNIPAm was set as a control group during the measurement.

The effect of the amount of organic solvents on the shape change was studied. And, finally, the fluorescence quenching experiment with Cy3-modified PS-embedded polymersomes with BHQ-modified PNIPAM was conducted to show that the fluorescence was quenched by the implementation of PNIPAm to the membrane in the presence of organic solvents. This work merits as a significant advance of the previously reported experiments, but fails to deliver enough new insights to warrant the publication in Nature Communications.

Answer: We thank the reviewer for the comments. We have recently published a paper in Nature Communications where PNIPAm addition method was used. However, we believe that does not decrease the novelty of this manuscript since here we focus in depth on the mechanism of new morphologies observed, including the octopus-like shapes, which researchers from various fields will appreciate. We believe the novelty of this manuscript is in its development of new and valuable dynamic morphologies with feasible and easy method and our focus on its mechanism of formation. The coronavirus challenge brought scientists together to develop nanocarriers for vaccine development with unprecedented speed. More and more attention has been placed onto efficient designs for delivery vehicles with controlled shapes, structures, and functions. Polymeric vesicles have become promising tools in the fields of micro/nanoreactors, drug delivery, and cell mimicking due to their enhanced membrane stiffness and chemical versatility in terms of flexibility and chemical design of the membrane and its properties. However, the scant shapes obtained from polymersomes restricted their full potential. For example, in nanomedicine, the shape of nanocarriers is considered a new parameter in governing their performance in vivo, as important as polymer property, particle size and surface chemistry. Therefore, shape control methods hold the promise to generate polymersomes with different morphologies and perform shape transformations on the bilayer membranes, which are not yet available nevertheless expected from scientists in different fields. This is particularly important since for a long time nanocarriers such as polymersomes were long considered inert spherical particles whose sole role is the encapsulation and transport of drugs. Here we challenge this believe and demonstrate the dynamic nature of the vesicles under environmental changes which is important to consider upon future applications. In the future, we hope this study can provide further understanding of the role of salt ions in biomembrane curvature formation and provide a functional toolbox for generation of exotic shapes of different morphologies that are not yet possible and available for nanomedicine applications.

In this manuscript, we have designed a new approach for the formation of multi-armed polymersomes by applying PNIPAm as a responsive hydrophobic unit and salt ions to modulate the property of PNIPAm and its interaction with the polymeric membrane. The use of salt ions is innovative and feasible. Polymersomes with multiple arms were fabricated, and the number of arms is salt concentration-dependent. Moreover, the mechanism behind this salt ions assisted PNIPAm insertion was studied by investigating the salt ion influence on PNIPAm and on PNIPAm-membrane interaction. This is the first time that the multi-armed polymersomes have been formed and studied, and also the first-time detailed information on curvature generation has been revealed in a membrane model. In the future, we hope this study can provide evidence of the role that salt ions play in biomembrane

curvature formation. This new strategy also allows us to fabricate polymer-based nano-systems with stimuli-responsive membranes for various applications. Our focus on the mechanism and the salt influence on shape changing could provide further evidence to support the role of salt ions in the formation of curvature in polymeric membranes and understand natural processes where salt ions and membrane proteins play crucial roles. Such insights are important for the future design of responsive and dynamic nanocarriers and will be appreciated by researchers from diverse backgrounds. Furthermore, in the nanomedicine field, nanocarriers such as polymersomes were considered inert spherical particles whose sole role is the encapsulation and transport of drugs. Here we challenge this believe and demonstrate the dynamic nature of the vesicles under environmental changes which is important to consider upon future applications.

Additional experiments with other salts in the Hofmeister series could clear the point that the presence of NaNO_3 is not only for imposing additional osmotic pressure, but also introduces specific interaction between PNIPAM/PEG and ions.

Answer: We thank the reviewer for the comment. The relevant experiments have been conducted and discussed in the manuscript. As shown below:

As shown in Supplementary Fig 5, we conducted control experiments to further explore the ion influence on membrane curvature formation with both the strong Hofmeister “kosmotropes” and “chaotropes”. Interestingly, strong “chaotropes” ions like NaSCN showed a positive influence in the construction of membrane curvature with an average of 11 arms formed, which is even slightly higher than NaNO_3 . Moreover, NaCl , with a similar Hofmeister effect as NaNO_3 , showed a similar impact on membrane curvature formation. On the other hand, when “kosmotropes” ions were added in the solution, the formation of membrane curvature is not as much evident as when NaNO_3 was added. Two “kosmotropes” ions Na_2HPO_4 and Na_2SO_4 , were added separately in these experiments, and both of them were found less effective than the “chaotropes”. Na_2SO_4 showed a much less influence in the formation of curvatures, 3-5 arms were formed on average. In general, “chaotropes” were found to penetrate the headgroup region of a hydrocarbon-packed monolayer and even lipid membranes since the presence of chaotropic salts in the subphase increased the surface pressure^{44,45}. We believe the chaotropic salts in our experiment had a similar effect on the polymeric membrane.

Supplementary Figure 5. Shape transformation of polymersomes with Hofmeister salt. 60 mM NaSCN, NaCl, NaNO₃, Na₂HPO₄, and Na₂SO₄ were added to polymersomes together with 250 μg of PNIPAm to explore the salt impact on membrane curvature formation. After 23% of organic solvent was added to the systems, samples were quenched and examined by TEM. Scale bar 1 μm.

Reviewer #2 (Remarks to the Author):

In the present paper, Sun et al. discuss the addition of poly(N-isopropylamide) (PNIPAm) as a means to control the shape change in polystyrene-poly(ethylene glycol) block copolymers. Namely, they look at the combination of PNIPAm with organic solvent, water and salt ions. They investigate the thermodynamics of the assembly by isothermal titration calorimetry.

The paper proposes a **fundamentally interesting concept for the idea of shape change**. However, this reviewer believes that many the great results obtained in this manuscript are somewhat overshadowed by the presentation of said results. The absence of chemical structures, the fact that acronyms not defined (e.g. PNIPAm and ITC in the abstract; PEG in the introduction), the absence of chemical structures and the dearth of figures make it very hard to follow even for people working in the field. In addition, the writing is not particularly concise and to-the-point, with lots of run-on sentences (see below) which detract from the results.

Answer: We thank the reviewer for appreciative comments regarding our findings and for considering our manuscript providing “fundamentally interesting concept for the idea of shape change”. We have revised accordingly the manuscript. The acronyms are defined and added in the manuscript. The chemical structures and their interactions are described in Figure3.

Specific comments:

In the introduction, "researchers was able to" should be replaced by "were able to" and there should be a comma after "stomatocyte-in-stomatocyte".

Answer: We thank the reviewer for the comment. The sentences have been corrected.

At the end of the first page, the opening clause "To mimic the protein interaction induced shape transformation in polymersomes" does not make sense. This reviewer suspects that there is a compound adjective somewhere. Do the authors mean protein-induced shape transformation? What interaction would cause the transformation?

Answer: We thank the reviewer for the comment. We indicate that some membrane proteins can insert into the lipid membrane and cause the shape transformation of cells. We have rephrased this sentence. As shown below:

To mimic the protein-membrane interaction induced shape transformation in cells, a new strategy has been developed recently in our group, in which poly(*N*-isopropylacrylamide) (PNIPAm) was used as alternatives for proteins with hydrophobic unit to insert into polymersome membrane²⁷.

The authors finally describe poly(*N*-isopropylamide) on page 3, it should not be capitalized and the *N* should be italicized.

Answer: We thank the reviewer for the comment. The mistake has been rectified.

Page 4, "the extra osmotic pressure added" should be revised to "the extra osmotic pressure experienced by". Saying that osmotic pressure is added seems strange.

Answer: We thank the reviewer for the comment. This sentence has been rephrased.

In the following sentence, this reviewer assumes that the authors mean "salt-ion-induced reduction". Otherwise, the sentence does not make sense. Generally, throughout the authors should pay attention to compound adjective rules (e.g. entropy-driven, salt-in, salt-out, etc.)

Answer: We thank the reviewer for the comments. The mistakes have been rectified.

The sentence starting with "Without salt ions" a couple lines below is a run on and is as such unintelligible. Why would polymersomes be transferred to flat disks? Do the authors mean that they will be transformed? The stomatocytes are half-opened not "half open".

Answer: We thank the reviewer for the comments. Here we mean "transformed", the polymersomes can be transformed to flat disks due to solvent exchange during the process. The sentence has been rephrased as shown below:

Without salt ions, polymersomes can be transformed to flat disks (Fig.1a1-a2, g). However, the addition of salt ions pushed the shape transformation further to half-open stomatocytes-like polymersomes (Fig. 1h) due to the extra inner volume reduction. It can be seen by the reduced distance between two bilayers from the EM images.

The TEM justification for the parametrization is not very strong. The authors show one image. The supporting information does not indicate an extensive analysis of the different polymersomes either. To be believable, the authors would need to perform said analysis on many micrographs and many assembled structures.

Answer: We thank the reviewer for the comments. We apologize for the unclear description. The fitted phase diagram is based on 5 assembled structures at each experiment condition. We did not perform a much bigger analysis since it is not that easy to collect a larger sample size and the volume reduction is very clear between groups.

Page 5, osmotic pressure cannot be given. Consider rephrasing.

The sentence starting with "Volume reduction has been considered" is a run-on sentence. Consider splitting it into two. The second fragment as written does not make sense.

The following sentence needs a reference to the previous work of the authors. The reader cannot be expected to know off the top of their heads the body of work of the authors.

Answer: We thank the reviewer for the comments. The sentences have been rephrased and the reference has been added as shown below:

When 1 mM, 3 mM, 15 mM and 60 mM NaNO₃ were added in the system, Δv decreased 49%, 62%, 64% and 66% respectively comparing to 0 mM salt (Fig. 1f). It indicates that adding salt ions can decrease the inner volume of polymersomes since the polymersome membrane has faced extra osmotic pressure³⁷. Besides, the volume reduction gets more difficult with the inner volume getting smaller since bending energy is higher at the membrane. Decreasing inner volume resulted in almost no difference in shape transformation. Although volume reduction has been considered the driving

force in the formation of structures like multi-armed starfish *etc* in liposome systems^{34,35,38}, the above results proved that tuning only the volume reduction will not lead to a further shape transformation. As studied in our previous work²⁷, membrane composition change was considered the critical condition for shape changing. We, therefore, assume that membrane composition change could be the main factor in the formation of arms.

A reference to the change in hydrophobicity of PNIPAm is needed. Also, the authors discuss the addition of NaNO₃ and then follow with the sentence: "This is based on the fact that PNIPAm can change its property from hydrophilic to hydrophobic when certain organic solvent is added." It is unclear how the addition of an inorganic compound is related to that. There is no apparent correlation at this point. The authors should indicate clearly what organic solvent they are adding.

Answer: We thank the reviewer for the comments. The sentences have been rephrased and the reference has been added as shown below:

As studied in our previous work²⁷, membrane composition change was considered the critical condition for shape changing. We, therefore, assume that membrane composition change could be the main factor in the formation of arms. In order to prove this hypothesis, 250 µg PNIPAm was added to the solution together with different amounts of NaNO₃. After 23% organic solvent was added to the solutions, samples were quenched and examined with TEM (Fig1.i1-n1) and Cryo-SEM (Fig1.i2-n2). Since PNIPAm can change its property from hydrophilic to hydrophobic when a certain amount of organic solvent (THF, Methanol, etc.) is added to its solution³⁹. We expect PNIPAm to insert into polymeric membranes as a hydrophobic motif to change the membrane composition based on hydrophobic effect²⁷. With the increase of salt concentration, polymersomes changed from boomerang-like shapes, tubes, and three armed shapes to shapes with multiple arms. The higher the salt concentration, the more arms formed on polymersomes. When salt concentration reached 15 mM, 6-7 long arms were observed. When salt concentration reached 30 mM, 7-8 short arms were observed, and the number of arms could be above 10 when 60 mM salt was added.

While this reviewer also tends to believe that these starfish structures are flat with protruding arms nothing prevents them from looking more like a nitrile glove that was inflated. Do the authors have additional proof of the flat nature of the center portion? Also, the authors should comment on the tip of the arms; Figure 1 m1 and n1 have especially interesting features.

Answer: We thank the reviewer for the comments. The inflated nitrile glove-like structures can also be achieved with different conditions. This manuscript will not discuss these results as the experimental conditions are different. Figure1m1 and n1 are TEM pictures; therefore, there is some

drying effect on the structures. That is why it is necessary to check the structures with cryo-SEM. With the cryo-SEM picture, more details can be observed. The flat nature of the center portion can be proved from cryo-SEM pictures taken from different angles, as shown below.

The authors discuss kosmotropes and chaotropes but this reviewer could not identify any experiments that demonstrated this effect here beyond the use of NaNO_3 . Is that an oversight? Is that a planned experiment? If so, it should appear in the conclusion rather than here.

Answer: We thank the reviewer for the comments. A relative experiment has been done to complete this discussion. As shown below:

As shown in Supplementary Fig 5, we conducted control experiments to further explore ion influence on membrane curvature formation with both the strong Hofmeister “kosmotropes” and “chaotropes”. Interestingly, strong “chaotropes” ions like NaSCN showed a positive influence in the construction of membrane curvature with an average of 11 arms formed, which is even slightly higher than NaNO_3 . Moreover, NaCl , with a similar Hofmeister effect as NaNO_3 , showed a similar impact on membrane curvature formation. On the other hand, when “kosmotropes” ions are added in the solution, the formation of membrane curvature is not as much as when NaNO_3 was added. Two “kosmotropes” ions Na_2HPO_4 and Na_2SO_4 , were added separately in these experiments, and both of them were found less effective than the “chaotropes”. Na_2SO_4 showed a much less influence in the formation of curvatures, 3-5 arms were formed on average. In general, “chaotropes” were found to penetrate the headgroup region of a hydrocarbon-packed monolayer and even lipid membranes since the presence of chaotropic salts in the subphase increased the surface pressure^{44,45}. We believe the chaotropic salts in our experiment had a similar effect on the polymeric membrane.

Supplementary Figure 5. Shape transformation of polymersomes with Hofmeister salt. 60 mM NaSCN, NaCl, NaNO₃, Na₂HPO₄ and Na₂SO₄ were added to polymersomes together with 250 μg of PNIPAm to explore the salt impact on membrane curvature formation. After 23% of organic solvent was added to the systems, samples were quenched and examined by TEM. Scale bar 1 μm.

As mentioned above with the organic solvent, the authors should define the co-nonsolvent used. Is that THF and dioxane? It is neither clearly mentioned in the text nor in the supplementary information. The sentence starting with "As a weak chaotrope" is a run-on. Consider splitting it into two. There are multiple instances where the authors start with "[long clause 1], however, [long clause 2]". Most of them can be split. Also, consider a different adverb to express contradiction than however to make it less monotonous as it is repeated many times (e.g. nevertheless, yet, although, whereas, energy-favoured...)

Answer: We thank the reviewer for the comments. The organic solvent we mentioned in this manuscript refers to THF and dioxane. We have made it clear in the manuscript, and the sentences have been rephrased accordingly.

The sentence indicating that "hydrophobic hydration was destroyed" will be puzzling to the non-specialist. The authors should define it, especially, as it is used later in the document.

Answer: We thank the reviewer for the comments. The hydrophobic hydration has been defined in the manuscript.

Above Figure 2, the authors discuss "the size and light scattering intensity". The authors need to explain what size they are referring to.

Answer: We thank the reviewer for the comments. We have made it clear in the manuscript, as shown below:

Furthermore, when the salt concentration increased to 30 and 60 mM, the size and light scattering intensity of these samples examined with DLS did not significantly change, meaning that the particles were not formed. Although the UV-Vis experiments proved the hydrophobicity of PNIPAm, DLS results reveal that PNIPAm chains remain to be separated due to the salt ions binding while turning hydrophobic (Fig. 2d), which facilitates the insertion of PNIPAm into the polymeric membrane.

Right below, the sentence starting with "However" is a run-on and is unintelligible.

Answer: We thank the reviewer for the comment. The sentences have been rephrased.

Figure 2, the caption title should also capture the variation in VFC. Figure 2b is presumably extracted from Figure 2c. This reviewer does not really understand the advantage of showing Figure 2b.

Figure 2d is not particularly clear. It is too small and does not really convey whatever the message the authors intend to pass onto the reader.

Answer: We thank the reviewer for the comments. We agree with the reviewer's suggestion and have made some changes to this figure. In addition, we made Figure 2e clear, and the meaning of each symbol has also been clarified as shown below:

Figure 2. The co-nonsolvency phenomenon of PNIPAm at different salt concentrations with VFC variate from 0% to 41%. (a) PNIPAm transmittance measurement by UV-vis spectroscopy to detect the shift of LCST with different salt concentrations (0 mM- 60 mM) at 23% of VFC. (b) The shift of LCST with VFC varies from 0% to 41%. (c) Dynamic light scattering measurements to observe the hydrophobic behaviour of PNIPAm at the VFC of 23% with the addition of 0, 1, 3, 15, 30, 60 mM NaNO₃ in PNIPAm water solution. (d) Chemical structure of PNIPAm and possible salt interaction position between PNIPAm and salt ions. (e) Schematic representation of hydrophobic behaviour of PNIPAm chains at the addition of salt ions and solvent, salt-only cannot turn PNIPAm to hydrophobic, but after addition of organic solvent (THF and dioxane), PNIPAm became hydrophobic, however, no interchain aggregation was observed.

While this reviewer appreciates the limitations of ITC, the assumption that temperature increase in PNIPAm is equivalent to solvent addition is bold. Is there some literature precedent that the authors can point to? Is there any calibration or experiment that can be done to demonstrate the veracity of this assertion?

Answer: We thank the reviewer for the constructive comments. As mentioned, the ITC has restrictions on the use of organic solvents. Therefore, we tried to think of a better way to explain our finding, and temperature increase is used as an alternative strategy in our experiment. Its theoretical basis is that lower critical solution temperature (LCST) of PNIPAm in water is believed to be strongly related to the destabilization of hydrogen bonds between water molecules and amide groups with increasing temperature, induced by the presence of the hydrophobic isopropyl group and backbone^{1,2}. The mixing process at low temperatures is favored by the formation of hydrogen bonds thermodynamically, which leads to a large negative enthalpy of mixing³. When a mixed solvent is added to the solution, PNIPAm solubility is reduced within a range of intermediate solvent concentrations in binary aqueous solutions. This phenomenon suggested that 'water-solvent complexes' are preferred to PNIPAm-water hydrogen bonds⁴. This indicates that both LCST changes share similar chemistry. Moreover, systematic research has been done to compare these changes⁵. Using IR and micro-Raman spectroscopy, the authors showed that gradual redshifts of the C-H stretching and the amide II bands happen with increasing temperature or DMSO concentration. Therefore, we believe our experiment result can help us and the readers to understand these

interactions behind the shape changes. We have added this reference in our manuscript to make the discussion more solid.

Page 14, hydrophobic instead of hydrophob/

Answer: We thank the reviewer for the comment. The mistake has been rectified.

Generally speaking, the explanation of the ITC results is both not thorough in terms of describing what is observed and what it means. This section needs to be improved upon for instance by providing actual number rather than qualitative explanation. The section also reads very much like a sequence of suppositions and the reader is left with the impression that the results are not fully understood.

Answer: We thank the reviewer for the constructive comments. We have revised this section of the manuscript. We hope the revised discussion is clear and easy to understand.

What do the authors mean by "flat but armed". "The same as previous observation" should be rephrased.

Answer: We thank the reviewer for the comments. Here we mean that these polymersomes have several arms, and at the same time, they are still flat disk structures. We corrected the sentence to make it more clear, as shown below:

Black crystal dots were observed along the membrane, as shown in Fig. 4a-b, meaning that PNIPAm is inserted all over the perimeter of these flat polymersomes with several arms. Similar to our previous observation²⁷, PNIPAm is inserted in the edge of the pancake-like shape where the curvature is higher so that there is enough space for the insertion to occur.

Page 16, "it's too close" should be "it is too close".

Answer: We thank the reviewer for the comments. We revised this sentence.

The demonstration about insertion via quenching should appear way earlier in the text. It is a bit out of place at the end.

Answer: We thank the reviewer for the comments. The insertion demonstration is also shown in Figure 3f. Here we mainly want to demonstrate the insertion with different amounts of organic solvent added during the shape transformation.

Overall, this work constitutes a highly worthy piece of work that should definitely be published. This reviewer however questions the novelty, especially, in light of a paper based on a similar platform by the authors in Nat. Comm. earlier this year. In addition, there are some minor technical questions that need to be addressed along with more important (yet manageable) writing issues.

Answer: We thank the reviewer for the comments. We have fully listened to your comments and have comprehensively revised this manuscript. Some other experiments have also been added accordingly. We hope that the revised article will meet your requirements, and further suggestions are appreciated.

Reviewer #3

On line 55 the reviewer found that "the protein-membrane intercation induced shape transformation" was still hard to understand, even after revision, and the same could be said pf other similar uses and phases.

Answer: We thank the reviewer for the comments. We have revised the sentence as showing below, we hope it is now more clear to the reviewer.

Membrane proteins can induce shape transformation in cells when interacting with biomembrane, in order to mimic this shape transformation, a new strategy has been developed recently in our group, in which poly(*N*-isopropylacrylamide) (PNIPAm) was used as alternatives for proteins with a hydrophobic unit to insert into polymersome membrane²⁷.

The reviewer felt that either fraction or percents needed to be used to express delta v on lines 114 and 115. The reviewer was of the opinion that many statements were not supported by reference to the data and figures (e.g. line 118/119, lines 136-138...) and requested more care be taken over this.

Answer: We thank the reviewer for the comments. We have revised the sentence as showing below, we hope it is now more clear to the reviewer.

When polymersomes were transformed to flat disks, Δv was reduced by 40%³⁶. When 1 mM, 3 mM, 15 mM and 60 mM NaNO₃ were added in the system, Δv decreased 49%, 62%, 64% and 66% respectively comparing to 0 mM salt (Fig. 1f). It indicates that adding salt ions can decrease the inner volume of polymersomes since the polymersome membrane has faced extra osmotic pressure³⁷. Besides, this osmotic pressure adds energy to the system which can stored on the polymersome membrane in the form of bending energy, results into deformation of the polymersome. For small inner volume polymersome, the high bending energy inhibits further volume reduction as more osmotic pressure is needed. (Supplementary Fig 1).

When salt concentration reached 15 mM, 6-7 long arms were observed (Fig1.i1 and i2). When salt concentration reached 30 mM, 7-8 short arms were observed (Fig1.m1 and m2), and the number of arms could be above 10 when 60 mM salt was added (Fig1.n1 and n2).

The reviewer also found some of the statements that referred to physical mechanisms to be just dropped in the text without being explained (e.g. 197-200 or 217-222).

Answer: We thank the reviewer for the comments. We have rephrased this part, we hope it is now more clear to the reviewer.

In order to investigate this possible interaction between salt and PNIPAm, we first evaluated the conformational change of PNIPAm by measuring solution transmittance using UV-vis spectroscopy at different NaNO₃ concentrations (0 mM - 60 mM) when 23% of volume fraction of co-nonsolvent (VFC) was added into the PNIPAm water solutions (Fig. 2a-b). The LCST of PNIPAm decreased when salt concentration increased from 0 mM to 15 mM. However, when more NaNO₃ was added into the system, LCST slightly rose back. The rise will be explained later. Another interesting observation is that the specific ion interaction with PNIPAm at 23% of VFC showcases a hydrophobic change beyond LCST, which means that even at low temperature, PNIPAm was slightly hydrophobic as long as salt is added into the system as seen from the decrease of transmittance in Fig. 2a. The percentage of transmittance at a lower temperature ($T < LCST$) is salt concentration dependent, the higher the salt concentration, the lower the transmittance is, which indicates that salt-induced conformational change in PNIPAm is different from an organic solvent. This has been studied in Previous research^{27,47,48}, the LCST of PNIPAm can be decreased by co-nonsolvency phenomenon, therefore with the change of solvent ratio, LCST changes. Simply put, the phase behaviour of PNIPAm in the water-rich and the alcohol-rich regime is different. Adding solvent to water decreases the enthalpy of the bulk water due to the kosmotropic effect. In this experiment, this effect suppresses the hydrophobic hydration between water and hydrophobic units of the PNIPAm while adding water to alcohol decreases the solubility of PNIPAm in the solvent mixture⁴⁹. On the other hand, as a weak "chaotrope", NaNO₃ can have a salt-in effect on PNIPAm as studied previously⁵⁰, meaning that they could stabilize PNIPAm by binding directly to the polyamide in a salt concentration dependent manner. Since the organic solvent is also involved in our study, PNIPAm is dispersed in the quaternary PNIPAM/water/solvent/salt system. In this system, both solvents and salt are agents that effectively affect the PNIPAm, assist each other to increase the surface tension of the cavity surrounding the backbone and the isopropyl side chains⁵⁰⁻⁵², so that the LCST slightly decreased as showing in Fig. 2a. When salt concentration is higher than 30 mM, the ions that break the hydrophobic hydration (water structure around hydrophobic molecules) are saturated

and extra ions might compete to bind on the side-chain amide moieties leading to a salt-in effect^{33,50,53}, which increases LCST (Fig. 2a). On the other hand, the cancelling of LCST could be due to the formation of a poor solvent system.

We have fully listened to your comments and have comprehensively revised this manuscript. Some other experiments have also been added accordingly. We hope that the revised article will meet your requirements, and further suggestions are appreciated.

- 1 Schild, H. G. & Tirrell, D. A. Microcalorimetric detection of lower critical solution temperatures in aqueous polymer solutions. *The Journal of Physical Chemistry* **94**, 4352-4356 (1990).
- 2 Ilmain, F., Tanaka, T. & Kokufuta, E. Volume transition in a gel driven by hydrogen bonding. *Nature* **349**, 400-401 (1991).
- 3 Walker, J. S. & Vause, C. A. Reappearing Phases. *Scientific American* **256**, 98-105 (1987).
- 4 Costa, R. O. R. & Freitas, R. F. S. Phase behavior of poly(N-isopropylacrylamide) in binary aqueous solutions. *Polymer* **43**, 5879-5885 (2002).
- 5 Yamauchi, H. & Maeda, Y. LCST and UCST Behavior of Poly(N-isopropylacrylamide) in DMSO/Water Mixed Solvents Studied by IR and Micro-Raman Spectroscopy. *The Journal of Physical Chemistry B* **111**, 12964-12968 (2007).

REVIEWER COMMENTS

Reviewer #4 (Remarks to the Author):

In this work, the authors investigated the morphological transition of polymersomes in the presence of PNIPAM upon adding salt and organic solvents. The authors have revised the manuscript, but several issues remain to be resolved.

1. In Figure 2, the authors used DLS to investigate the size changes at varying salt concentrations. It appeared that in some cases (at high salt concentrations) the PNIPMA solution had rather low transmittance. The reviewer was doubtful whether the DLS method was reliable in determining the sizes in these cases. The same problem can be found in Figure 5I.
2. In Figure 2d, it should be PNIPAM chain rather than PNIPAM chain.
3. The interaction between PNIPAM and PEO chains was important to drive the morphological transitions. However, the evidence provided in Figure 4C (EDS mapping) result was of poor quality. The reviewer was wondering if it was possible to use the FRET technique to investigate this process.
4. TEM results shown in Figure 5a-h were good. However, SEM results shown in Figure 5a1-h1 was not clear enough. What were the objects in the background?

Reviewer #5 (Remarks to the Author):

I have been asked to assess if the the technical issues raised by Reviewer #1 and Reviewer #2 have been addressed adequately.

Reviewer 1:

I can confirm that the technical issues raised by Reviewer 1 have been adequately addressed.

Reviewer 2:

- The reply to the comment: "The TEM justification for the parametrization is not very strong. The authors show one image. The supporting information does not indicate an extensive analysis of the different polymersomes either. To be believable, the authors would need to perform said analysis on many micrographs and many assembled structures." by Reviewer 2 seems to not have been implemented in the manuscript. I would recommend to mention that five images have been used.
- Caption of supplementary Figure 5 mentions addition of "23% of organic solvent". This needs to be specified further in the caption: which organic solvent was added. Secondly, what does 23% refer to, i.e. what is 100%?
- Figure 2d: there is a typo in the figure, which says "Chian" but probably should read "chain".
- Figure 2d: I don't understand where the ions interact with the PNIPAm.
- It is unclear whether the (sufficient) answer to the question "While this reviewer appreciates the limitations of ITC, the assumption that temperature increase in PNIPAm is equivalent to solvent addition is bold. Is there some literature precedent that the authors can point to? Is there any calibration or experiment that can be done to demonstrate the veracity of this assertion?" has been implemented in

the manuscript. It should be added as it is useful for the broad readership of Nature Communications.

General comment:

I would like to mention that the authors could have improved on the clarity of their response, for example by highlighting not only by color their reply but by addition of the word “reply” at the beginning of each paragraph, and by numbering the different requests. It was also not always clear from the rebuttal if something had been included in the manuscript as in part outlined above. Generally, I agree with Reviewer 2 some additional effort could have been made in writing the manuscript clearly.

REVIEWER COMMENTS

Reviewer #4 (Remarks to the Author):

In this work, the authors investigated the morphological transition of polymersomes in the presence of PNIPAM upon adding salt and organic solvents. The authors have revised the manuscript, but several issues remain to be resolved.

1. In Figure 2, the authors used DLS to investigate the size changes at varying salt concentrations. It appeared that in some cases (at high salt concentrations) the PNIPMA solution had rather low transmittance. The reviewer was doubtful whether the DLS method was reliable in determining the sizes in these cases. The same problem can be found in Figure 5l.

Reply: We thank the reviewer for the comments. Transmittance refers to the fraction of incident light that passes through a sample without being absorbed or scattered. It is related to the concentration, thickness, and optical properties of the sample, as well as the wavelength of the incident light. In the DLS measurement, the concentration we used is 20 times lower than the UV-vis measurement, so it should not be influenced by the lower transmittance. On the other hand, the DLS measurement is based on light scattering, which means it is more related to the particle size. We think the DLS measurement is reliable.

2. In Figure 2d, it should be PNIPAM chain rather than PNIPAM chian.

Reply: We thank the reviewer for the comment. The mistake has been corrected.

3. The interaction between PNIPAM and PEO chains was important to drive the morphological transitions. However, the evidence provided in Figure 4C (EDS mapping) result was of poor quality. The reviewer was wondering if it was possible to use the FRET technique to investigate this process.

Reply: We thank the reviewer for the comment. Indeed, the resolution of EDS mapping cannot go higher, but nevertheless can clearly demonstrate that PNIPAm is present near the membrane. In addition, PNIPAm has to be able to attach on the membrane, otherwise it would be removed by centrifugation during the experiment. FRET technique was used in our previous paper to prove the insertion (reference 27), and from there we have learned that the PNIPAm is inserted into the membrane. In Fig 5i, the FRET technique is also used to follow the deep insertion of PNIPAm.

4. TEM results shown in Figure 5a-h were good. However, SEM results shown in Figure 5a1-h1 was not clear enough. What were the objects in the background?

Reply: We thank the reviewer for the comment. The pictures in Figure 5a1-h1 are cryo-SEM pictures. Sample preparation was required before the images were taken and the background is normal and related to the cryoSEM technique. The particles were dispersed in water and quickly frozen with liquid nitrogen or liquid helium to prevent the formation of ice crystals. Before imaging, the sample is fractured by a sharp knife followed by etching to reveal the objects on the surface. When we scan this fractured surface, which is freeze etched to reveal the objects, we can see the particles at the interface, as shown in the image.

Combining the TEM images (a-h), we can see a1 is a spherical structure, b1 is rather flat, c1 has an opening, and the tentacles are spherical-like structures. From d1 to g1, the tentacles slowly disappear. We can confirm that the object in the background is just vitrified water or fractured vitrified water, which is not visible in the other technique. In different regions, the

vitrified water may show different morphology, it is normal to see this when taking cryo-SEM pictures because the cut of the fracture may occur at different angles (Figure 5 is attached below).

Reviewer #5 (Remarks to the Author):

I have been asked to assess if the technical issues raised by Reviewer #1 and Reviewer #2 have been addressed adequately.

Reviewer 1:

I can confirm that the technical issues raised by Reviewer 1 have been adequately addressed.

Reviewer 2:

1. The reply to the comment: "The TEM justification for the parametrization is not very strong. The authors show one image. The supporting information does not indicate an extensive analysis of the different polymersomes either. To be believable, the authors would need to perform said analysis on many micrographs and many assembled structures." by Reviewer 2 seems to not have been implemented in the manuscript. I would recommend to mention that five images have been used.

Reply: We thank the reviewer for the comment. The 5 images used in the analysis has been attached in Supplementary Figure 1, and the calculations are added as Supplementary Table 1. And the data has been adapted in the manuscript.

Supplementary Figure 1. (b) Cryo-TEM images of the 5 structures used for parametrization from each salt concentration.

Supplementary Table 1. Parameterization of different shapes from cryo-TEM images (n=5). For every fitted shape, the reduced volume and area (Δv) are calculated.

1mM	Sample 1	Sample 2	Sample 3	Sample 4	Sample 5	Average	SD
ΔV	0.25	0.33	0.26	0.33	0.27	0.288	0.039
Δa	0.36	0.38	0.33	0.49	0.4	0.392	0.061
3mM	Sample 1	Sample 2	Sample 3	Sample 4	Sample 5		
ΔV	0.28	0.24	0.24	0.21	0.24	0.242	0.025
Δa	0.55	0.45	0.45	0.58	0.45	0.496	0.064

15mM	Sample 1	Sample 2	Sample 3	Sample 4	Sample 5		
ΔV	0.22	0.22	0.18	0.23	0.22	0.214	0.019
Δa	0.64	0.34	0.5	0.54	0.34	0.472	0.131

60mM	Sample 1	Sample 2	Sample 3	Sample 4	Sample 5		
DV	0.19	0.18	0.21	0.15	0.23	0.192	0.030
DA	0.42	0.45	0.44	0.48	0.37	0.432	0.041

2. Caption of supplementary Figure 5 mentions addition of "23% of organic solvent". This needs to be specified further in the caption: which organic solvent was added. Secondly, what does 23% refer to, i.e. what is 100%?

Reply: We thank the reviewer for the comment. The details have been added, and the caption has been rephrased as shown below:

After solvent mixture (THF:Dioxane =4:1) was added to the system to reach a concentration of 23% (v/v) organic solvent, samples were quenched and examined by TEM. Scale bar 1 μm .

3. Figure 2d: there is a typo in the figure, which says "Chian" but probably should read "chain".

Reply: We thank the reviewer for the comment. The mistakes have been corrected.

4. Figure 2d: I don't understand where the ions interact with the PNIPAm.

Reply: We thank the reviewer for the comment. When Salt is added to PNIPAm solution, anion (nitrate ions) can directly bind to the amide group of PNIPAm, causing the destabilization through polarization. We made it clear by adding the polarization of PNIPAm. The explanation is also adapted in the manuscript, as shown below:

(d) Chemical structure of PNIPAm and possible salt interaction position between PNIPAm and salt ions. When Salt is added to PNIPAm solution, anion (nitrate ions) can directly bind to the amide group of PNIPAm, causing the destabilization through polarization.

5. It is unclear whether the (sufficient) answer to the question "While this reviewer appreciates the limitations of ITC, the assumption that temperature increase in PNIPAm is equivalent to solvent addition is bold. Is there some literature precedent that the authors can point to? Is there any calibration or experiment that can be done to demonstrate the veracity of this assertion?" has been implemented in the manuscript. It should be added as it is useful for the broad readership of Nature Communications.

Reply: We thank the reviewer for the comment. We also agree that this discussion should be implemented in the manuscript, and it has been added, as shown below:

The ITC has restrictions on the use of organic solvents. Therefore, temperature increase is used as an alternative strategy in our experiment. Its theoretical basis is that the lower critical solution temperature (LCST) of PNIPAm in water is believed to be strongly related to the destabilization of hydrogen bonds between water molecules and amide groups with increasing temperature, induced by the presence of the hydrophobic isopropyl group and backbone 54,55.

The solvent mixing process at low temperatures is favored by the formation of hydrogen bonds thermodynamically, which leads to a large negative enthalpy of mixing⁵⁶. When a mixed solvent is added to the solution, PNIPAm solubility is reduced within a range of intermediate solvent concentrations in binary aqueous solutions. This phenomenon suggested that 'water-solvent complexes' are preferred to PNIPAm-water hydrogen bonds⁵⁷. This indicates that both LCST changes share similar chemistry. Moreover, systematic research has been done to compare these changes⁵⁸. Using IR and micro-Raman spectroscopy, the authors showed that gradual redshifts of the C-H stretching, and the amide II bands happen with increasing temperature or DMSO concentration. We therefore believe ITC can be a good approach to assist further the investigation.

6. General comment:

I would like to mention that the authors could have improved on the clarity of their response, for example by highlighting not only by color their reply but by addition of the word "reply" at the beginning of each paragraph, and by numbering the different requests. It was also not always clear from the rebuttal if something had been included in the manuscript as in part outlined above. Generally, I agree with Reviewer 2 some additional effort could have been made in writing the manuscript clearly.

Reply: We thank the reviewer for the comment. We have numbered the comments and started with "reply" with all the answers, all the changes made in the manuscript are also attached below the related comments with a different color. We have also made some revisions in the manuscript to make it clearer. The changes can be checked in the manuscript with review tracking.

REVIEWERS' COMMENTS

Reviewer #4 (Remarks to the Author):

The authors have properly revised the manuscript according to the reviewers' comments. The revised manuscript is now publishable.

Reviewer #5 (Remarks to the Author):

The comments have now been sufficiently addressed and implemented in the manuscript. I thank the authors for the clearly structured reply.